# Genomic insights into the probiotic potential and genes linked to gallic acid metabolism in *Pediococcus pentosaceus* MBBL6 isolated from healthy cow milk

**Md. Morshedur Rahman**[1], **Naim Siddique**[1], **Soharth Hasnat**[1,2], **Md. Tanvir Rahman**[3], **Mustafizur Rahman**[4], **Munirul Alam**[4], **Ziban Chandra Das**[1], **Tofazzal Islam**[2]*, **M. Nazmul Hoque**[1]*

**1** Department of Gynecology, Obstetrics and Reproductive Health, Molecular Biology and Bioinformatics Laboratory, Bangabandhu Sheikh Mujibur Rahman Agricultural University, Gazipur, Bangladesh, **2** Institute of Biotechnology and Genetic Engineering (IBGE), Bangabandhu Sheikh Mujibur Rahman Agricultural University, Gazipur, Bangladesh, **3** Faculty of Veterinary Sciences, Department of Microbiology and Hygiene, Bangladesh Agricultural University, Mymensingh, Bangladesh, **4** iccdr'b (International Centre for Diarrhoeal Disease Research, Bangladesh), Dhaka, Bangladesh

* tofazzalislam@bsmrau.edu.bd (TI); nazmul90@bsmrau.edu.bd (MNH)

**Data Availability Statement:** The genome sequenced and analyzed during the current study is available at NCBI under BioProject PRJNA1065156.

## Abstract

*Pediococcus pentosaceus* is well known for its probiotic properties, including roles in improving health, antimicrobial production, and enhancing fermented food quality. This study aimed to comprehensively analyze the whole genome of *P. pentosaceus* MBBL6, isolated from healthy cow milk, to assess its probiotic and antimicrobial potentials. *P. pentosaceus* MBBL6, isolated from a healthy cow milk at BSMRAU dairy farm, Gazipur, Bangladesh, underwent comprehensive genomic analysis, including whole genome sequencing, assembly, annotation, phylogenetic comparison, and assessment of metabolic pathways and secondary metabolites. Antimicrobial efficacy was evaluated through *in-vitro* and *in-vivo* studies, alongside *in-silico* exploration for potential mastitis therapy. We predicted 1,906 genes and 204 SEED sub-systems involved in carbohydrate metabolism and vitamin B complex biosynthesis, with a focus on lactose metabolism in MMBL6. Notably, 43 putative carbohydrate-active enzyme genes, including lysozymes, suggest the ability of MBBL6 for carbohydrate biotransformation and antimicrobial activity. The genome also revealed primary metabolic pathways for arginine and gallic acid metabolism and secondary metabolite gene clusters, including T3PKS and RiPP-like regions. Importantly, two bacteriocin biosynthesis gene clusters namely *bovicin_255*_variant and *penocin_A*, were identified in MBBL6. The safety assessment of MBBL6 genome revealed no virulence genes and a low pathogenicity score (0.196 out of 1.0). Several genes related to survival in gastrointestinal tract and colonization were also identified. Furthermore, MBBL6 exhibited susceptibility to a wide range of antibiotics *in-vitro*, and effectively suppressed mastitis pathogens in an *in-vivo* mouse mastitis model trial. The observed bacteriocin, particularly bovicin, demonstrated the ability to disrupt the function of an essential protein, Rho factor of mastitis pathogens by blocking transcription termination process. Taken together, our in-depth genomic

The associated Sequence Read Archive (SRA) and BioSample accession number are SRR27558660 and SRS20151026, respectively.

**Funding:** This work was supported by research grants from the BSMRAU Innovation Fund 2023-2024 (Grant No. 14) and the Research Management Wing (RMW), BSMRAU (Grant No. 18, FY 2023-2025), Bangladesh. The funders had no role in study design, data collection and analysis, decision to publish, or preparation of the manuscript.

**Competing interests:** The authors have declared that no competing interests exist.

analysis underscores the metabolic versatility, safety profile, and antimicrobial potential of *P. pentosaceus* MBBL6, suggesting its promise for applications in therapeutics, bioremediation, and biopreservation.

## 1. Background

Probiotics are living or viable beneficial microorganisms, such as bacteria and yeast, that provide health benefits to the hosts, when administered in adequate amounts [1]. Over the last two decades, probiotics have gained more attention for conferring extensive beneficial effects without causing any harms to the host health [1,2]. Lactic acid bacteria (LAB) are considered as the most prevalent and beneficial probiotics among the various bacterial genera. The LAB comprises a considerable number of bacterial species belonging to genera *Lactobacillus*, *Lactococcus*, *Leuconostoc*, *Pediococcus*, *Bifidobacteria* and *Streptococcus* [3]. These microorganisms are extensively used due to their ability to ferment carbohydrates into lactic acid, which helps maintain a healthy gut microbiome [1]. Historically, LAB have been intensively used in food fermentation as natural fermenters. The U.S. Food and Drug Administration (FDA) categorizes LAB as "Generally Recognized As Safe" (GRAS), while the European Food Safety Authority (EFSA) designates them under the Qualified Presumption of Safety (QPS) status [4]. These classifications highlight the health benefits of LAB, which include enhancing digestion, boosting the immune system, and potentially preventing gastrointestinal infections [1,5]. Thus, the extensive study of LABare essential to further underscore their critical role in promoting overall health and well-being.

*Pediococcus pentosaceus* is a widely distributed and well-recognized probiotic LAB species commonly found in milk and milk products, fermented foods, meat, feces, the gut, plants, and plant products [5,6]. It has been reported that *P. pentosaceus* can survive in diverse environments, including animal and human guts, supported by genes that facilitate adaptation, stress tolerance, and resistance [7]. These genetic traits, including those associated with surface adhesion, enable *P. pentosaceus* to attach to various substrates and host tissues, essential for colonization and survival across ecological niches [8]. Moreover, *P. pentosaceus* is involved in several metabolic processes such as carbohydrates, lipids, amino acids, energy, and vitamins metabolism [5,7]. *P. pentosaceus* is also recognized for producing antimicrobial compounds like pediocin, a bacteriocin with broad-spectrum activity against bacterial pathogens, including *Escherichia coli*, *Staphylococcus aureus*, *Bacillus subtilis*, *Klebsiella pneumoniae*, *Pseudomonas aeruginosa*, *Enterococcus faecalis*, *Acinetobacter baumannii*, as well as *Listeria* and *Clostridium* species [9–13]. These bacteriocins are significant in food preservation, combating spoilage, preventing foodborne diseases, and show promise as therapeutic agents against clinical and foodborne illnesses [14–16]. Furthermore, strains of *P. pentosaceus* have demonstrated potential health benefits including antiobesity, antiinflammatory, antioxidant, antimicrobial, and cholesterol-lowering effects, along with enhancing intestinal barrier function and modulating the immune system [14,16,17]. Gallic acid (GA), a naturally occurring bioactive phenolic compound presents in a variety of plants and fruits like bananas, lemons, pineapples, grapes, and strawberries [18]. GA and its derivatives are known for conferring numerous beneficial effects on host health, such as antioxidant, antimicrobial, anticancer, anti-inflammatory, gastroprotective, cardioprotective, neuroprotective, and metabolic disease prevention activities [18,19]. To provide health benefits, phenolic compound must be bioavailable in the gastrointestinal tract. Certain LAB species possess genes involved in GA metabolism, which can enhance its bioavailability and significantly amplify its beneficial effects in the human body [20,21].

Advances in next-generation sequencing technology have expanded the array of tools available for investigating genomic insights into microorganisms [22,23]. For instance, whole genome sequencing offers a comprehensive view of genetic variations within individuals [23,24]. Furthermore, genomic analysis of probiotics to identify antimicrobial resistance (AMR) and virulence genes provides crucial insights into potential risks associated with their future use [13]. Additionally, genome sequencing reveals insights into the functional diversity, metabolic pathways, antibiotic character, and health benefits of probiotics [1,2]. This study aimed to sequence and analyze the genome of *P. pentosaceus* MBBL6, isolated from healthy cow's milk, to elucidate its genomic characteristics, safety profile, antimicrobial efficacy, and probiotic potential (**S1 Graphical** abstract). Findings of this study revealed that *P. pentosaceus* MBBL6 possesses useful metabolic pathways, strong antimicrobial properties, and excellent safety, making it a promising probiotic for preventing mastitis in dairy farms and other biotechnological applications.

## 2. Materials and methods

### 2.1 Ethics statement

This study was reviewed and approved by the Animal Research Ethics Committee (AREC) of the Bangabandhu Sheikh Mujibur Rahman Agricultural University, Bangladesh (Reference number: FVMAS/AREC/2023/6679, Date: 16/01/2023).

### 2.2 Bacterial strain and culture conditions

Milk samples from healthy lactating cows were collected from the BSMRAU dairy farm (24.09° N, 90.41° E) in Gazipur, Bangladesh. To confirm the absence of mastitis, a California Mastitis Test was performed [25]. The milk samples were then enriched in deMan, Rogosa and Sharpe (MRS) broth (HiMedia, India) and incubated overnight aerobically at 37°C. The enriched samples were streaked onto MRS agar plates and incubated at 37°C for 48 h. The MBBL6 isolate was identified as *P. pentosaceus* using the VITEK-2 system v9.01 [26]. Pure colonies from MRS agar were diluted in 0.45% saline to achieve a 0.5 McFarland standard and subsequently inoculated into VITEK 2 system cards. Within 3 h, the system identified the MBBL6 isolate as *P. pentosaceus* based on their distinctive biochemical properties. Finally, *P. pentosaceus* MBBL6 isolate was stored in MRS broth at -80°C supplemented with 20% glycerol [6].

### 2.3 Genome sequencing, assembly and annotation of the *P. pentosaceus* MBBL6

The MBBL6 isolate was subcultured overnight at 37°C in nutrient broth (Biolife™, Italy). Genomic DNA extraction was performed using the QIAamp DNA Mini Kit (QIAGEN, Germany). The purity and concentration of the extracted DNA were assessed using a NanoDrop 2000 spectrophotometer (Thermo Fisher Scientific, USA) [27]. Whole genome libraries were prepared from 1 ng of DNA using the Nextera™ DNA Flex library preparation kit (Illumina, San Diego, USA) and subjected to whole genome sequencing (WGS) on an Illumina MiSeq sequencer (Illumina, USA) with a 2 × 250 bp protocol [6]. Paired-end raw reads (N = 4,216,200) were trimmed with Trimmomatic v0.39 [28] and quality-checked using FastQC v0.11.7 [29]. Reads with phred scores > 20 [30] were assembled with SPAdes v3.15.6 [31]. Genome annotation was performed using the NCBI Prokaryotic Genome Annotation Pipeline (PGAP) v6.6 [32], utilizing the *Pediococcus* CheckM v1.2.2 marker set [33]. Furthermore, Rapid Annotation using Subsystem Technology (RAST, http://rast.nmpdr.org/) [34] online web server was used for subsystem analysis. The MBBL6 genome was visualized using

CGView (http://stothard.afns.ualberta.ca/cgview_server/; accessed 20 June 2024). The tRNAscan-SE v2.0 [35] web search server (http://trna.ucsc.edu/software/; accessed 20 June 2024), Barrnap software (https://github.com/tseemann/barrnap) and PlasmidFinder [36] were used to detect tRNAs, rRNAs and plasmid regions in the assembled genome of MBBL6, respectively. Furthermore, subsystem features and metabolic function-related genes/pathways of MBBL6 were identified using RAST server v2.0 [37]. Genome annotation and classification of metabolic pathways utilized the KEGG database [38] via KEGG Automatic Annotation Server (KAAS, https://www.genome.jp/kegg/kaas/).

## 2.4 Average nucleotide identity (ANI) and phylogenetic analysis

The assembled MBBL6 genome was used to calculate ANI values and to conduct evolutionary phylogenetic analysis with nineteen closely related *P. pentosaceus* strains (**S1 Table**) available in the National Center for Biotechnology Information (NCBI) database (https://www.ncbi.nlm.nih.gov/). ANI values were calculated for all-vs-all microbial genomes using ANIclustermap v.1.3.0 [39]. Multiple alignments of nucleotide sequences of the assembled draft genomes (**S1 Table**) were performed with MUSCLE v3.8 [40], and a maximum-likelihood (ML) phylogenetic tree was generated using PhyML v3.3 [41]. The tree was then visualized using iTOL v6.9 [42].

## 2.5 Comparative genome analysis

To detect orthologous gene groups among twenty *P. pentosaceus* strains (**S1 Table**), including the MBBL6 strain, OrthoFinder v2.5.5 [43] was used with default settings. The UpsetR package [44] in R v4.3.3 was utilized to create an Upset plot for visualizing orthogroups identified by OrthoFinder. Additionally, the core orthogroups were classified into Clusters of Orthologous Groups (COG) [45] families using the Batch CD-Search web server (https://www.ncbi.nlm.nih.gov/Structure/bwrpsb/bwrpsb.cgi; accessed 01 July 2024). The genomes of *P. pentosaceus* DSPZPP1 and SRCM217654, which clustered into the same clade as MBBL6 in the phylogenetic tree, were selected for further analysis of the potential evolutionary relationships among these strains. To demonstrate synteny between large blocks of genomic sequences, the Mauve alignment tool [46] was applied. Additionally, the BLAST Ring Image Generator (BRIG) v0.95 [47] was used to compute and visualize the nucleotide sequences of *P. pentosaceus* strains MBBL6, DSPZPP1, MBBL4, and SRCM217654. The *P. pentosaceus* SRCM217654 strain served as the standard (reference) genome for comparative analysis.

## 2.6 Analysis of carbohydrate utilization and associated enzymes in *P. pentosaceus* MBBL6

A series of biochemical tests (**S2 Table**) were conducted following standard protocols [48] to evaluate the carbohydrate fermentation profiles and enzyme activities of the MBBL6 isolate. To examine the carbohydrate metabolism of *P. pentosaceus* MBBL6, the Carbohydrate Active Enzyme Database (CAZy, http://www.cazy.org/) was used to investigate the genes associated with CAZymes families. The protein sequences of the genome were annotated with HMMER v3.4 [49] via dbCAN server (https://bcb.unl.edu/dbCAN2/index.php) against the CAZy database. To identify the CAZyme class, the cut-off values of $1e^{-15}$ and 0.35 were used.

## 2.7 Analysis of primary metabolic regions and secondary metabolites in *P. pentosaceus* MBBL6

In MBBL6 genome, primary metabolic gene clusters were detected using gutSMASH v1.0 [50] and secondary metabolite biosynthetic gene clusters were predicted with antiSMASH v7.0

[51]. Data analysis and visualization were conducted using the SRplot and the R v4.3.3 with the packages (ggplot2, dplyr, and viridis) [52]. The BAGEL4 (http://bagel4.molgenrug.nl/index.php; accessed 05 July 2024) web tool was applied to identify the bacteriocins and their associated genes present in the MBBL6 genome. Additionally, the BLASTp (https://blast.ncbi.nlm.nih.gov/Blast.cgi) was used to manually confirm the identified bacteriocin domains against databases of non-redundant protein sequences (nr).

## 2.8 *In-vitro* and genomic safety analyses of *P. pentosaceus* MBBL6

The antibiotic susceptibility test for the MBBL6 isolate was conducted using the Kirby-Bauer disc diffusion method [23]. The susceptibility pattern was assessed with ten commercially available antibiotic discs, including those inhibiting protein synthesis (azithromycin, 15 μg; amikacin, 30 μg; and gentamicin, 10 μg), cell wall synthesis (cefepime, 30 μg; ceftriaxone, 30 μg; cefuroxime sodium, 30 μg; imipenem, 10 μg; and meropenem, 10 μg), and DNA gyrase inhibitors (nalidixic acid, 30 μg; and ciprofloxacin, 5 μg) (HiMedia). In brief, overnight cultures of approximately $10^9$ CFU/mL were swabbed onto Mueller Hinton agar plates. Antibiotic-impregnated discs were then placed on the swabbed plates under aseptic conditions. After 24 h of incubation at 37˚C, the diameters of the inhibition zones were measured. The tests were performed in triplicate, and the results were classified as resistant (R), intermediate (I), or susceptible (S) according to the Clinical and Laboratory Standards Institute (CLSI) M100 guidelines of 2023 [53]. Antibiotic resistant genes (ARGs) were predicted using the ResFinder 4.0 [54] database. Further, virulence genes were searched using VirulenceFinder 2.0 [55] database. All the analysis was performed in the ABRicate (https://github.com/tseemann/abricate) biotool with default parameter. The PathogenFinder (http://cge.cbs.dtu.dk/services/PathogenFinder/, accessed 10 July 2024) online tool was utilized for calculating the possibility of the pathogen being human. The hemolytic activity of MBBL6 was evaluated by streaking the isolate on blood agar plates supplemented with 5% (v/v) sheep blood. After incubating the plates at 37˚C for 24 h, they were inspected for specific hemolytic reactions (α, β, or γ hemolysis) [56]. Control plates without the bacterial inoculum were used simultaneously for comparison. The PHASTEST (Phage Search Tool with Enhanced Sequence Translation) [57] web tool (https://phastest.ca/; accessed 02 July 2024) was used to search prophages in MBBL6 genome. The clustered regularly interspaced short palindromic repeats (CRISPR) and their CRISPR-associated (Cas) proteins were investigated using CRISPRCasFinder v4.2.20 [58] and insertion sequences (IS) of the genome were predict via ISfinder [59].

## 2.9 *In-vitro* and *in-vivo* antimicrobial efficacy of the *P. pentosaceus* MBBL6

The MBBL6 strain was evaluated for its probiotic potential against two major pathogens of bovine mastitis [60] such as *S. aureus* D4C4 (NCBI GenBank accession: MN620430) and *E. coli* G1C5 (NCBI GenBank accession: MN620426), using both *in-vitro* (agar well diffusion technique) and *in-vivo* (mouse mastitis model) methods. In brief, agar well diffusion method was performed using concentrated cell-free supernatant (CFS) of MBBL6 following established protocols with minor adjustments [13,61]. Initially, MBBL6 was cultured in 10 mL of nutrient broth (Oxoid, UK) and statically incubated at 37˚C for 24 h. Subsequently, 1 mL of this culture was transferred to 100 mL of nutrient broth and further incubated at 37˚C for 48 h. The resulting culture was then centrifuged at 6000 rpm for 15 min at 4˚C, followed by neutralization to pH 7.0 and sterile filtration through a 0.22 μm Millipore filter to obtain the CSF. Thereafter, inoculums ($10^6$ CFU/mL) of *S. aureus* and *E. coli* were spread on nutrient agar plates (Oxoid, UK). Several wells (6 mm in diameter) were made in each nutrient agar plate and filled with

100 μL of CFS from MBBL6. Following a 24 h incubation period at 37°C, the antibacterial activity was assessed by measuring the diameter of the inhibition zones.

To investigate the antimicrobial effectiveness of MBBL6 *in-vivo*, we conducted a mouse mastitis model experiment using 20 germ-free Swiss albino timed pregnant mice (Day 18 of breeding), divided into two groups: Group I (n = 10) and Group II (n = 10), following our previously published protocol [23,62]. The mice were obtained from the Animal Research Farm, International Centre for Diarrhoeal Disease Research, Bangladesh (ICDDR'B), Bangladesh. Mastitis was induced in both groups by subcutaneous injection (in the inguinal regions) and oral administration of *S. aureus* and *E. coli* inoculate at 100 μL per mouse ($10^6$ CFU/mL) [23]. After seven days of challenge, Group I mice were treated with 100 μL of MBBL6 ($10^9$ CFU/mL) administered orally and sprayed on the mammary gland every alternate day for 14 days. Group II mice received a placebo treatment with distilled water (control). The mice were housed in a germ-free environment, maintained on a 12 h light/dark cycle with ad libitum access to food and water. Their mammary glands were closely monitored each day for any signs of mastitis. Key clinical indicators of mastitis included noticeable redness, swelling, and discoloration of the mammary glands, along with the presence of exudate from the inguinal glands, signaling infection and inflammation. At the end of the experiment (*i.e.*, 14 days post-challenge), mice from each group were placed in a $CO_2$ chamber for euthanasia and to collect samples [63]. Specifically, to minimize suffering, euthanasia was performed humanely using $CO_2$ gas, followed by cervical dislocation for collection of mammary and colon tissues to assess inflammation levels and examine histological changes [64]. Histopathological sections of mammary and colon tissues were prepared using hematoxylin and eosin staining, following our established protocols [23]. Histopathological changes in the mammary and colon tissues were graded on a scale from 0 to 3 (0: normal, 1: mild, 2: moderate, 3: severe) [23,63]. Slides were examined using a Primo Star microscope (Carl Zeiss, Germany) with a 40× objective lens. Images were captured with an AxioCamERc5s camera through ZEN 2012 INK software (Carl Zeiss, Germany) and analyzed using Image J software.

## 2.10 *In-silico* therapeutic effect of *P. pentosaceus* MBBL6 against mastitis pathogens

The Pan-genome Explorer was used to analyze two complete genomes of *S. aureus* D4C4 and *E. coli* G1C5 sequenced from bovine clinical mastitis milk to identify common core proteins shared between the two species [65]. Essential proteins were identified from these core proteins by comparing them to a database of essential proteins using the Manual BLASTp program [66]. Bacteriocins were then screened against these essential proteins using the Maestro BioLuminate (v.5.5) software package from Schrödinger [67]. For final validation of the bacteriocins (e.g., bovicin and penocin) effects on the essential proteins of *S. aureus* D4C4 and *E. coli* G1C5, a 200 ns molecular dynamics simulation was conducted using the Maestro Desmond (v.14) software package from Schrödinger [67].

## 3. Results

### 3.1 *P. pentosaceus* MBBL6 exhibits typical genomic features

The draft genome of MBBL6 comprises 1,849,397 base pairs (bp) spanning over 42 contigs, with a genome coverage of 128.7x, and a GC content of 37.3%, indicating robust sequencing depth and a balanced nucleotide composition. It possessed two linear plasmid regions containing repUS64 (954 bp) and rep31 (1,158 bp). **Fig 1** demonstrates the circular view of the MBBL6 genome. The CheckM analysis of the genome revealed a high completeness of 97.97% and a low contamination level of 4.68%. The basic information of the MBBL6 genome and its genomic features, including

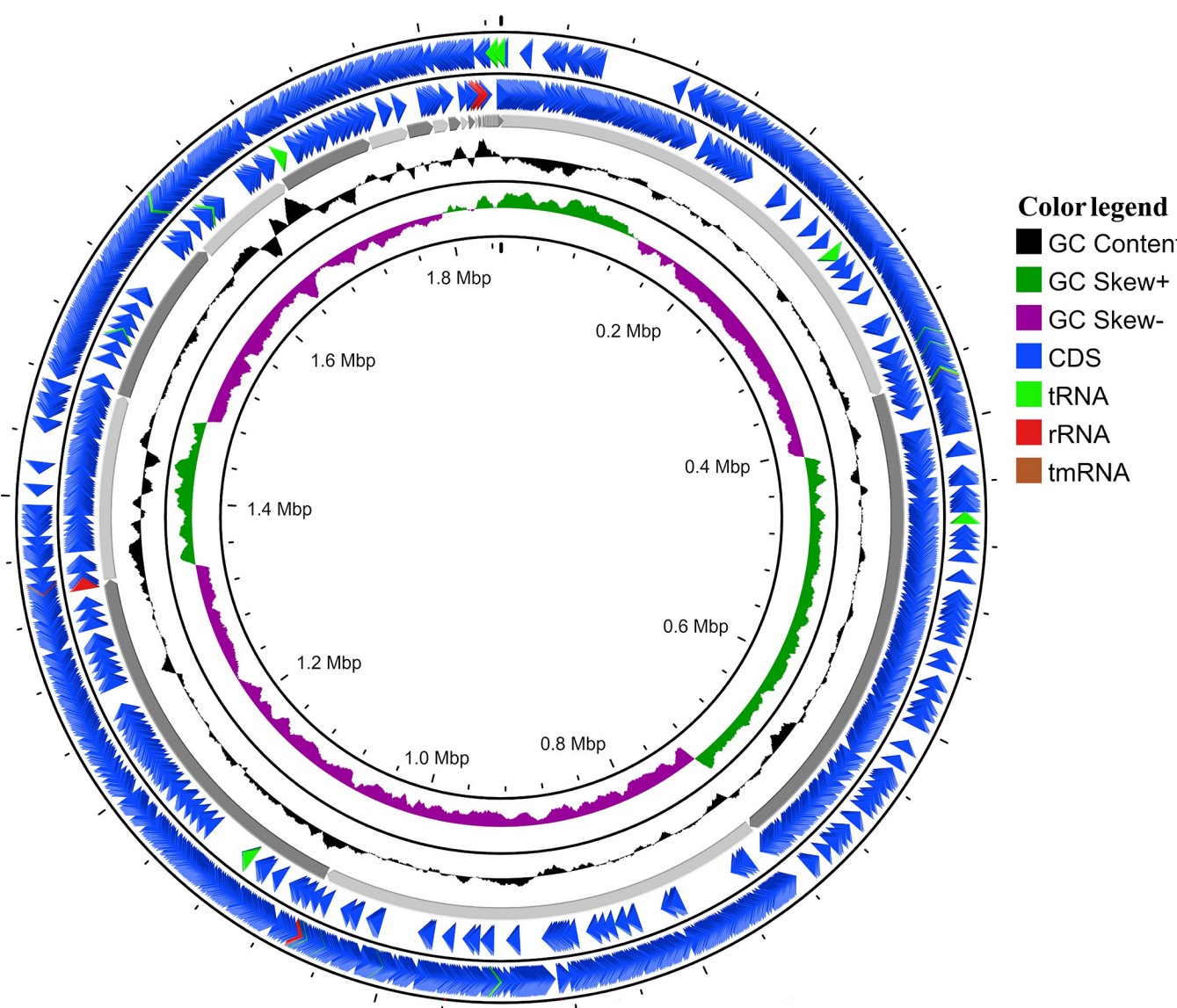

**Fig 1. Circular genome map of *P. pentosaceus* MBBL6.** The circular illustration was visualized using the CGView server and contains four rings. The outermost circle and the second circle show the positions of the coding sequences (CDSs) in forward and reverse strand directions, respectively. The tRNA, tmRNA e rRNA genes are represented by light green, red, and maroon arrows, respectively. The next rings show the following information: GC content (black), GC skew + (dark green), and GC skew – (purple), respectively.

coding sequences (CDSs), CRISPR regions, prophages, ARGs, virulence genes, and probability of being a human pathogen, are provided in **Table 1**. Genome annotation revealed 1,906 predicted genes, including 1,804 protein-coding sequences (CDS), two ARGs viz. *lnu*(A) and *erm*(B), seven rRNA genes (**Table 1**), and 55 tRNA genes (**S3 Table**). Additionally, 204 subsystems for metabolic functions were identified via RAST [34] annotation web server (**Table 1**).

## 3.2 *P. pentosaceus* MBBL6 reveals close phylogenetic relationship with other *P. pentosaceus* strains

The average nucleotide identity (ANI) of the MBBL6 genome was evaluated against nineteen other closely related genomes (**S1 Table**) available in the NCBI database. The ANIclustermap of these genomes revealed that all *P. pentosaceus* strains share at least 98% similarity with the

**Table 1. Genomic features of *P. pentosaceus* MBBL6.**

| Features | MBBL6 genome |
|---|---|
| Genome size (bp) | 1,849,397 |
| No. of contigs | 42 |
| Genome coverage (x) | 128.7 |
| GC content (%) | 37.3 |
| CheckM completeness (%) | 97.97 (4.68% contamination) |
| $N_{50}$ value (bp) | 333,724 |
| Total genes | 1,906 |
| Protein coding genes | 1,804 |
| tRNA | 55 |
| rRNA | 07 |
| tmRNA | 01 |
| Plasmid region (linear) | 02 |
| CRISPER | 03 |
| Cas protein | 0 |
| Prophages | 02 intake, 01 incomplete, and 1 questionable |
| ARGs | 02 [*lnu*(A), *erm*(B)] |
| Virulence gene | 0 |
| Pathogenic families matched | 0 |
| Probability of being human pathogen | 0.169 |

MBBL6 strain (**S1 Fig**). For instance, *P. pentosaceus* MBBL4 exhibited higher ANI (99.5%) score with MBBL6 genome, followed by *P. pentosaceus* C2OK (98.6%), *P. pentosaceus* SRCM217654 (98.6%), and *P. pentosaceus* SPARC2 (98.6%). The ANI inferences were further supported by the findings from the phylogenetic analysis. The maximum-likelihood phylogenetic tree (**Fig 2**) revealed that *P. pentosaceus* strains MBBL4, DSPZPP1, and SRCM217654 were clustered in the same clade with our sequenced genome of MBBL6.

### 3.3 *P. pentosaceus* MBBL6 shares core functional orthogroups with other *P. pentosaceus* strains

A comparative genomic analysis was performed between MBBL6 and 19 closely related *P. pentosaceus* strains to identify genetic similarities and differences, validate phylogenetic relationships, and gain insights into the functional capabilities and potential adaptations of MBBL6. During homologous gene analysis of twenty *P. pentosaceus* strains, a total of 35,799 genes were identified. Of these, 35,211 genes (98.4%) were grouped into 2,423 orthogroups using OrthoFinder [43]. Within these orthogroups, MBBL6 was present in 1,728 (71.3%) orthogroups, with only one orthogroup specific to a particular species. Additionally, among these orthogroups, 1,223 were identified as core orthogroups, along with 1,200 accessory orthogroups (**Fig 3A**). Furthermore, out of the 1,223 core orthogroups analyzed, 972 genes (79.48%) were classified into 23 functional COG categories (**Fig 3B**). The most prevalent category among these core orthogroups was "J—translation, ribosomal structure, and biogenesis" (12.15%), while the least prevalent was "Z—cytoskeleton" (0.9%).

### 3.4 *P. pentosaceus* MBBL6 exhibits minimal genomic discrepancy with reference genomes

For a comprehensive visualization of the coding sequences, the genome of MBBL6 was aligned with two strains of *P. pentosaceus*. In this analysis, MBBL6 genome showed high level of

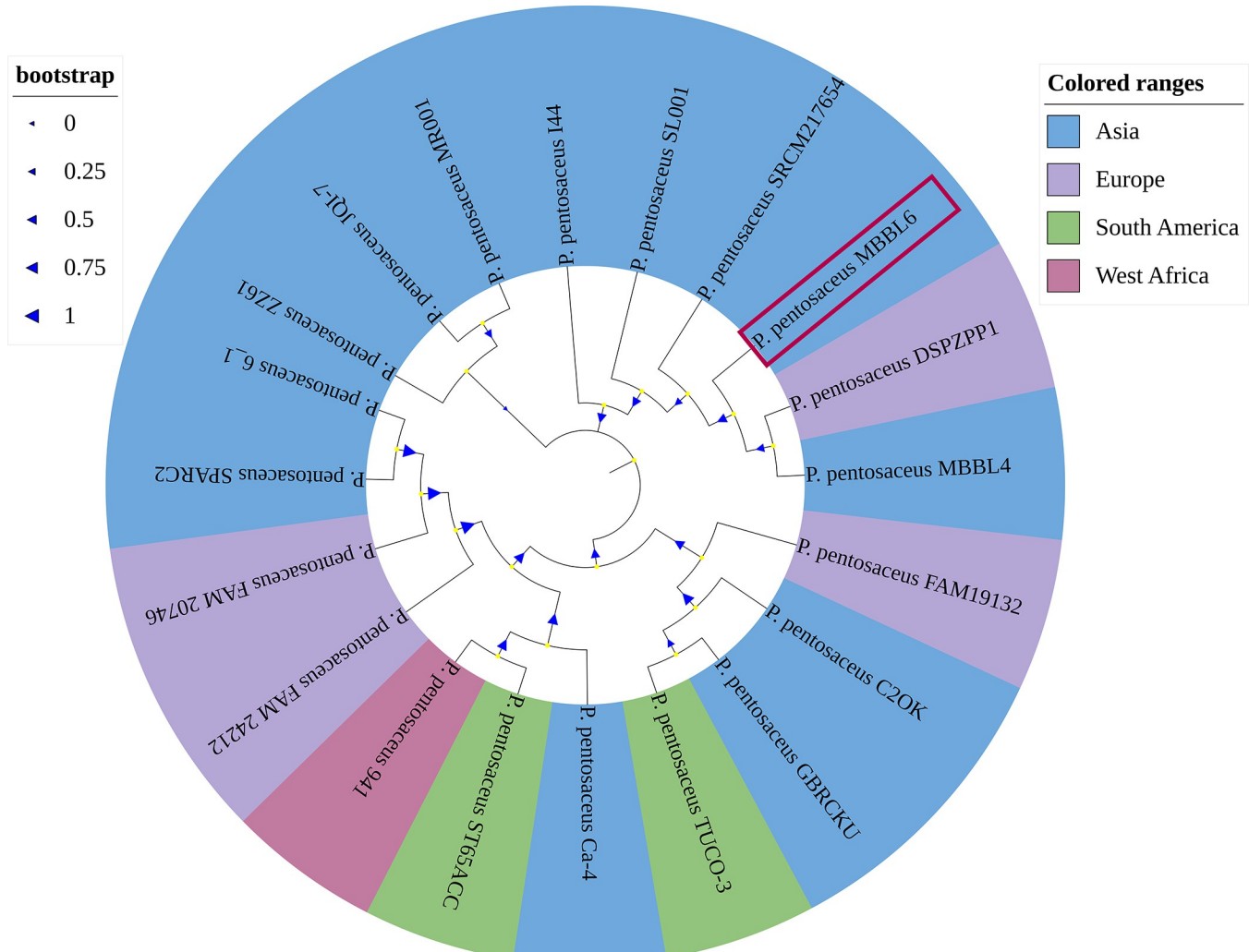

**Fig 2. The evolutionary relationships between *P. pentosaceus* MBBL6 and 19 reference genomes of *P. pentosaceus* strains sequenced from four continents worldwide.** Whole genome sequences of 19 strains of *P. pentosaceus* isolated from milk of lactating mammals (cow and cat), fermented food products, environmental sample, human and animal gut, cow feces, shrimp, and uterus of healthy buffalo origin retrieved from the NCBI were used for phylogenetic analysis. The mid-point rooted tree was constructed using the NCBI Tree Viewer (https://www.ncbi.nlm.nih.gov/tools/treeviewer/, accessed: June 20, 2024), and visualized with iTOL (interactive tree of life, accessed: June 20, 2024). The evolutionary relationship was inferred using the maximum-likelihood method. Different colors (e.g., light blue for Asia, pale purple for Europe, light green for South America, and light pink for West Africa) are assigned according to the close evolutionary relatedness (clade) of the genomes. The values on the branches are bootstrap support values based on 1000 replications. The genome of the *P. pentosaceus* MBBL6 (JAZIFR000000000) is highlighted on red box.

synteny with *P. pentosaceus* DSPZPP1 and SRCM217654 genomes (**Fig 4A**) isolated from traditional fermented sausages and cow feces, respectively. All three genomes shared a significant number of similarity regions. However, the genes were not found in the same positions and, in some instances, were placed in inverted positions within each chromosome. Additionally, some similarity regions were detected in only two of the three genomes. For example, the coral-colored region (position: 1011296) was found in both the MBBL6 genome (position: 1011296) and the SRCM217654 genome (position: 941585), but it was absent in the DSPZPP1 genome. The absence of such regions in some genomes suggests that even within the same sspecies, significant genomic divergence can occur. In addition, a comparative visualization

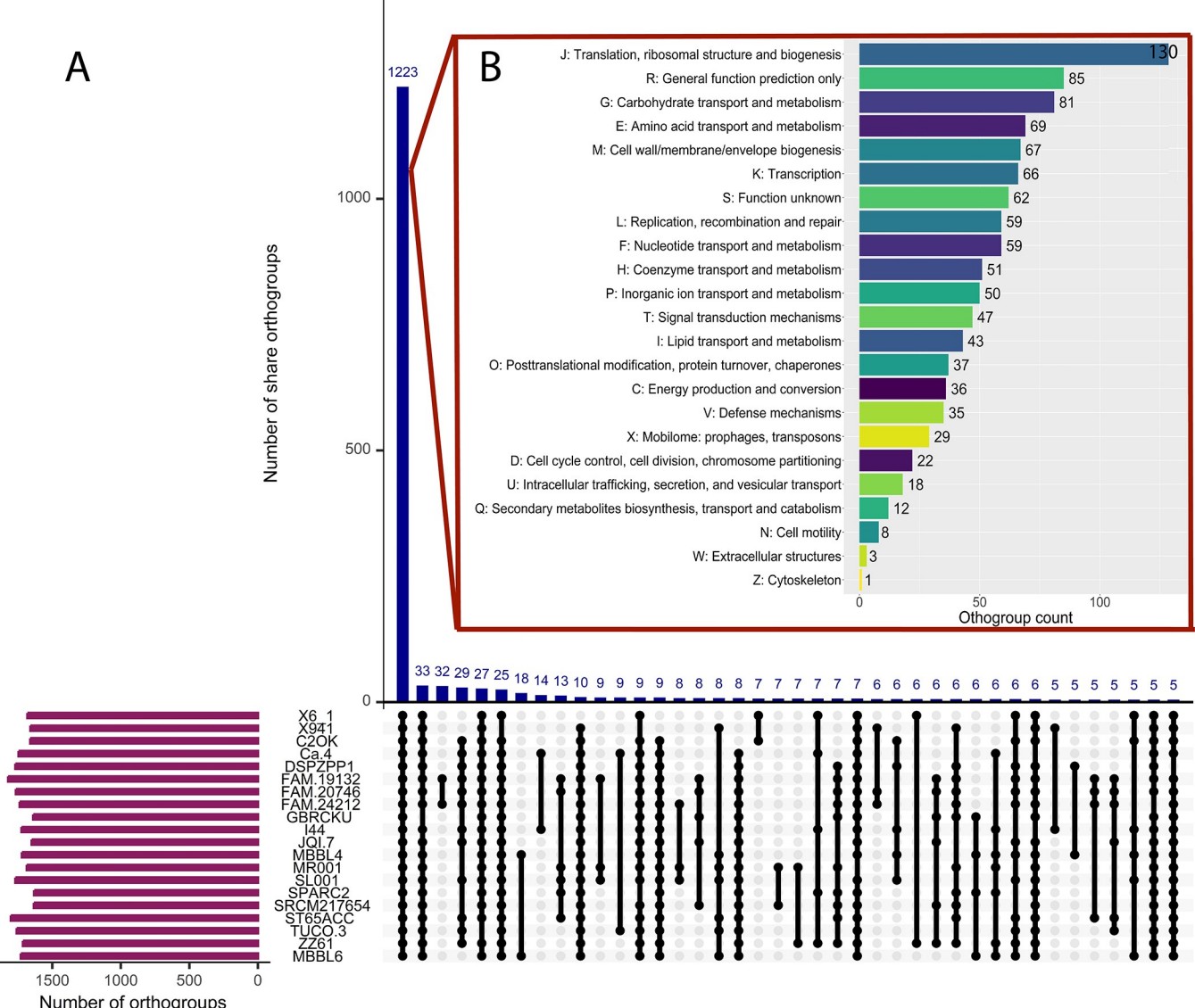

**Fig 3.** (A) Upset plot comparing shared orthogroups among the 20 *P. pentosaceus* species. The box plot displays the distributions of mean orthogroup protein lengths. Blue bar plots represent the number of orthogroups with specific conservation patterns, with numbers provided above each bar. (B) The bar chart with different colors shows the functional categories of core orthogroups based on COG classification, using the Batch CD-Search server (https://structure.ncbi.nlm.nih.gov/Structure/bwrpsb/bwrpsb.cgi, accessed: June 20, 2024).

(**Fig 4B**) was performed using the BRIG tool with the four *P. pentosaceus* strains presented in the same clade in the phylogenetic tree (**Fig 2**). In the BRIG genome circle (**Fig 4B**), similarities between the genomes were displayed by the solid sections within circle, and variation was represented by the blank sections. The BRIG comparison of the genomic maps indicated minimal large-scale variation among the bacterial genome sequences. Despite this, a significant number of homologous regions were observed around the reference genome, showing more than 95% identity. The comparative results also demonstrated that the reference genome (*P. pentosaceus* SRCM217654) has an open reading frame but not in the other genomes, which may be correlated with variation in assembly quality [5].

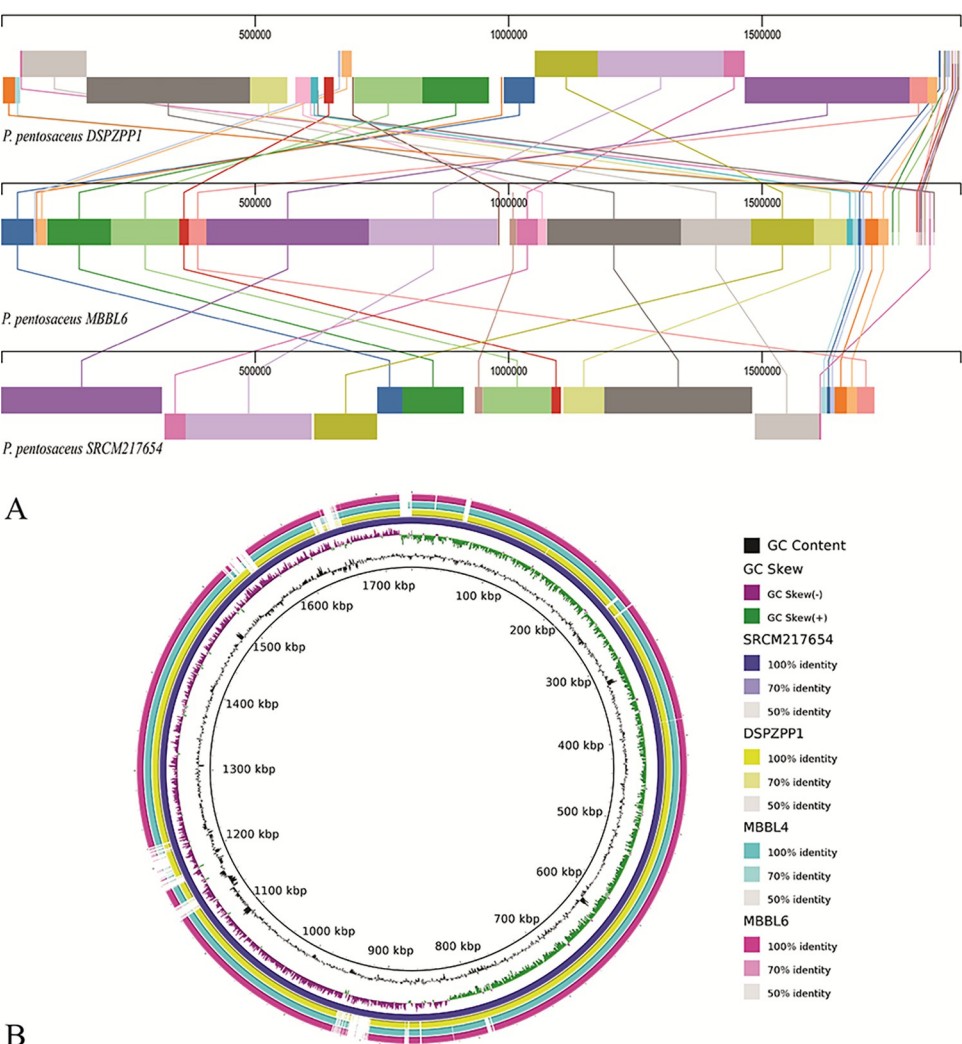

**Fig 4. Multiple genome alignment of *P. pentosaceus* strains.** (A) Multiple genome alignment of *P. pentosaceus* DSZPP1, *P. pentosaceus* MBBL6, and *P. pentosaceus* SRCM217654. Visualization of alignment is organized into one horizontal panel per genome sequence, with the label of the genome sequence name at the left of each panel. The homologous blocks are represented with the same color and are connected within the genome by lines. (B) Nucleotide alignments of four *P. pentosaceus* genomes, generated with the BLASTn with an e-value cut-off 1e-5 using BRIG 0.95. Circles (from inside to outside) 1 and 2 (GC content; black line and GC skew; purple and green lines), circle 3 (reference *P. pentosaceus* SRCM217654 genome; dark blue color), circle 4 (mapped *P. pentosaceus* DSPZPP1 genome; yellow color), circle 5 (mapped *P. pentosaceus* MBBL4 genome; cyan color), and circle 6 (mapped *P. pentosaceus* MBBL6 genome; magenta color).

## 3.5 *P. pentosaceus* MBBL6 utilizes diverse metabolic pathways to facilitate adaptation and survival

We further performed pathway analysis to gain a comprehensive understanding of how MBBL6 utilizes nutrients, interacts with its environment, and potentially influences health or disease processes. In MBBL6, a total of 977 genes (52.31%) were assigned to KEGG pathways, categorized into 38 sub-categories within six major functional categories. Among these, the highest proportion of genes were assigned to carbohydrate metabolism (13.51%), followed by translation (8.09%), replication and repair (6.65%), and amino acid metabolism (6.55% (**Fig 5A**). For a better understanding, metabolic pathways related to carbohydrate, amino acid,

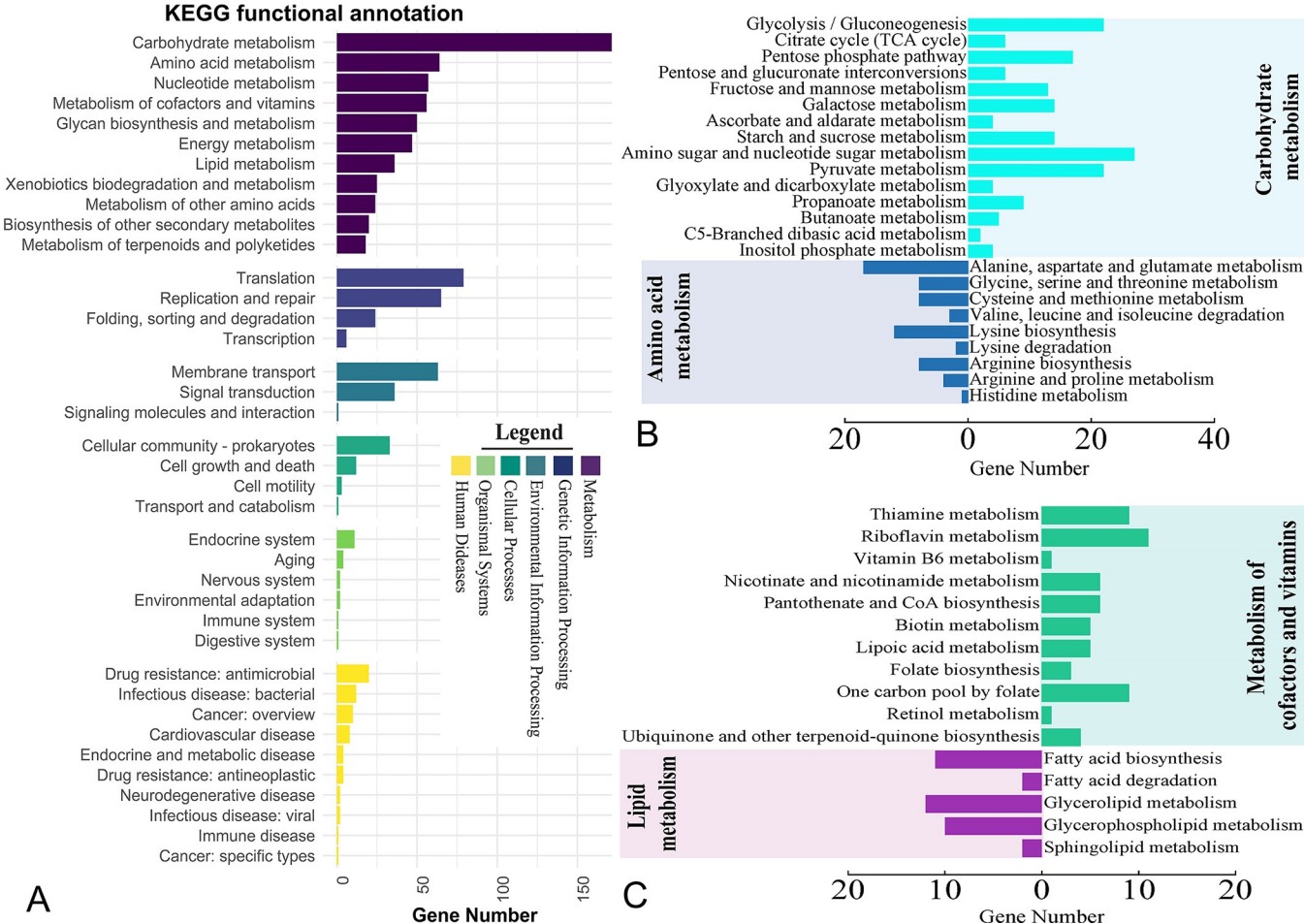

**Fig 5. Metabolic functional potentials of the *P. pentosaceus* MBBL6 annotated through the Kyoto Encyclopedia of Genes and Genomes (KEGG).** (A) Detailed representation of the functional classes belonging to six main functional categories. Each functional category is represented by different colors as follows: Dark purple for metabolism, navy blue for genetic information processing, Ocean blue for environmental information processing, teal color for cellular processing, light green color for organismal systems, and yellow color for human diseases. (B) Subcategories of carbohydrate and amino acid metabolism are represented by cyan and blue color bar charts, respectively. (C) Subcategories of cofactors and vitamins metabolism, and lipid metabolism are represented by mint and purple color bar charts, respectively.

lipid, and vitamin subcategories are illustrated in **Fig 5B** and **5C**. By analyzing the KEGG metabolic pathways we found that MBBL6 genome contained complete set of genes for glucose (e.g., *glk*, *gapA*, *gpmA*, *eno* etc.), galactose (e.g., *galM*, *galK*, *galT* etc.) and pentose phosphate (e.g., *pfkA*, *rbsK*, *prsA*, *rpiA* etc.) metabolism pathways (**S4 Table**). Additionally, two lactose metabolism related genes *lacZ* (beta-galactosidase) and *lacD* (tagatose 1,6-diphosphate aldolase) were found in MBBL6 genome. The Leloir metabolic pathway related protein encoded genes, which are considered for the catabolism of D-galactose in uridine diphosphate (UDP) glucose, were also found in this genome; *galM* (aldose 1-epimerase), *galK* (galactokinase), *galE* (UDP-glucose 4-epimerase), and *galT* (UDP glucose—hexose-1-phosphate uridylyl-transferase). Further analysis revealed that the MBBL6 strain also contains genes involved in pyruvate metabolism (**S4 Table**). These genes enable the conversion of pyruvate to acetyl-CoA through the pyruvate dehydrogenase complex (*pdhABCD*), and subsequently to acetate via phosphate acetyltransferase (*pta*) and acetate kinase (*ackA*). In the presence of oxygen, pyruvate can alternatively be converted to carbon dioxide and diacetyl phosphate by pyruvate oxidase (*poxL*).

**Table 2. Genes involved in the biosynthesis of vitamin B2 (riboflavin) of *P. pentosaceus* MBBL6 genome.**

| Gene ID | KO ID | Gene | Functions |
|---|---|---|---|
| gene-V4W90_RS02370 | K00793 | *ribE, RIB5* | Riboflavin synthase [EC:2.5.1.9] |
| gene-V4W90_RS02380 | K00794 | *ribH, RIB4* | 6,7-dimethyl-8-ribityllumazine synthase [EC:2.5.1.78] |
| gene-V4W90_RS08435 | K00878 | *thiM* | Hydroxyethylthiazole kinase [EC:2.7.1.50] |
| gene-V4W90_RS07935 | K01515 | *nudF* | ADP-ribose diphosphatase [EC:3.6.1.13 3.6.1.-] |
| gene-V4W90_RS02365 | K11752 | *ribD* | Diaminohydroxyphosphoribosylaminopyrimidine deaminase/5-amino-6-(5-phosphoribosylamino) uracil reductase [EC:3.5.4.26 1.1.1.193] |
| gene-V4W90_RS01005 | K11753 | *ribF* | Riboflavin kinase/FMN adenylyl transferase [EC:2.7.1.26 2.7.7.2] |
| gene-V4W90_RS02375 | K14652 | *ribBA* | 3,4-dihydroxy 2-butanone 4-phosphate synthase/GTP cyclohydrolase II [EC:4.1.99.12 3.5.4.25] |
| gene-V4W90_RS07685 | K19285 | *nfrA1* | FMN reductase (NADPH) [EC:1.5.1.38] |
| gene-V4W90_RS00910 | K21064 | *ycsE, yitU, ywtE* | 5-amino-6-(5-phospho-D-ribitylamino) uracil phosphatase [EC:3.1.3.104] |
| gene-V4W90_RS06550 | K20861 | *ybjI* | FMN hydrolase/5-amino-6-(5-phospho-D-ribitylamino) uracil phosphatase [EC:3.1.3.102 3.1.3.104] |
| gene-V4W90_RS02445 | K20861 | *ybjI* | FMN hydrolase/5-amino-6-(5-phospho-D-ribitylamino) uracil phosphatase [EC:3.1.3.102 3.1.3.104] |
| gene-V4W90_RS06850 | K03186 | *ubiX, bsdB, PAD1* | Flavin prenyltransferase [EC:2.5.1.129] |

Additionally, the MBBL6 genome includes the gene for acetate kinase (*ackA*), which facilitates the conversion of pyruvate to acetate. Although the genome lacks complete pathways for amino acid synthesis, it does contain the gene encoding aminopeptidase N (*pepN*) (**S4 Table**).

In addition, the assembled genome of MBBL6 harbored several genes associated with probiotics, including those encoding heat shock and stress proteins (*hsp20*, *groEL*, *dnak*, *recA*, *clpB*, *clpX*, *clpP*), surface adhesion genes (*ltaS*, *tuf*, *yidC*), acid tolerant genes (*atpABCDEFGH*, *celA*, *celB*, *celC*), antioxidant genes (*trxA*, *trxB*, *dapB*, *gor*, *zwf*, *gntZ*, *msrC*), folic acid biosynthesis genes (*folA*, *folC*), and immune gene (*metK*). The clandestine proteins encoding gene such as *gapA* (glyceraldehyde-3-phosphate dehydrogenase), and surface-associated gene *eno* (enolase) were also identified in the MBBL6 genome. The probiotic-related genes, along with their KO IDs, predicted in the MBBL6 genome, can be found in **S5 Table**. Furthermore, annotation of the sequenced genome using the KEGG KAAS online server identified numerous genes involved in amino acid (alanine, glycine, serine, threonine, cysteine, valine, leucine, lysine, arginine, histidine, tyrosine, tryptophan, phenylalanine) and vitamin B-complex (thiamine, riboflavin, nicotinamide, pantothenic acid, pyridoxine, biotin, folate) biosynthesis (**S5 Table**). Further analysis revealed that the MBBL6 strain harbors several genes associated with vitamin B-complex biosynthesis. Notably, genes such as *ribA/B/D/E/F/H*, *thiM*, *ybjI*, *nfrA1*, *ubiX* etc. identified in MBBL6, were found to be involved in the riboflavin biosynthesis pathway (M00125) (**Table 2**, **Fig 6**).

### 3.6 *P. pentosaceus* MBBL6 demonstrates capabilities for carbohydrate utilization and various enzymatic activities

The isolated MBBL6 strain demonstrated the ability to ferment several carbohydrates, including maltose, mannitol, mannose, sucrose, trehalose, and N-acetyl glucosamine (**S2 Table**).

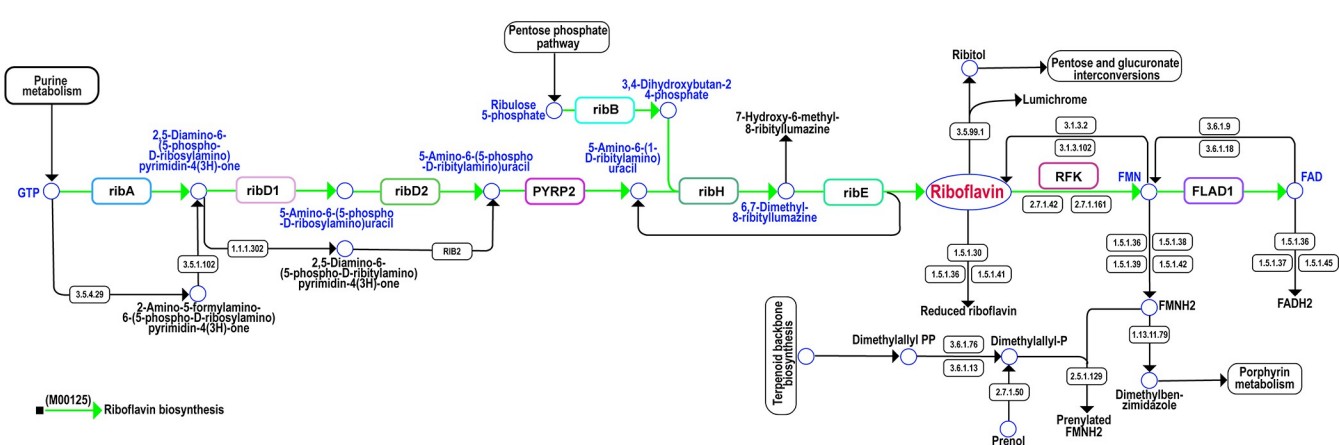

**Fig 6. Riboflavin biosynthesis pathways in *P. pentosaceus* MBBL6 genome.** The upper flowchart shows the riboflavin byosynthesis pathways (M00125) in MBBL6 genome in light green color with red line.

Enzymatic assays revealed positive activities for arginine utilization, urease production, and beta-glucosidase activity. Conversely, the strain tested negative for alkaline phosphatase, beta-glucuronidase, and beta-galactosidase activities. The Kligler Iron Agar (KIA) test showed a uniform yellow coloration in both the slant and butt, indicating acid production. Additionally, the strain was negative for indole production, methyl red reaction, Voges-Proskauer reaction, and citrate utilization (S2 Table). Furthermore, the MBBL6 genome was investigated against the CAZY database and forty-three genes were identified across four CAZymes classes, including glycosyltransferase (GT), glycoside hydrolase (GH), carbohydrate esterase (CE), and auxillary activity (AA) family (Fig 7). Among the major CAZymes classes, GT (20 genes) was identified as the most prevalent CAZymes family, followed by GH (18 genes), CE (4 genes), and AA (1 gene).

### 3.7 *P. pentosaceus* MBBL6 possesses gene repertoire for key primary metabolic regions and secondary metabolites

Studying primary metabolic regions is crucial for understanding cellular function and energy production, while secondary metabolites, such as bacteriocins, play a significant role in developing natural antimicrobials, probiotics, and tailored formulations, thereby advancing food safety, health, agriculture, and ecological insights. Two primary metabolic regions (Fig 8A and 8B, S6 Table) and secondary metabolite regions (Fig 8C and 8D, S6 Table) were identified within the genome of MBBL6 through the gutSMASH [50] and antiSMASH [51] server, respectively. The primary metabolic gene clusters in the MBBL6 genome, such as arginine2_h-carbonate and gallic_acid_met (gallic acid metabolism), exhibited exact similarity (100%) with arginine to hydrogen carbonate in *P. aeruginosa* and gallic acid degradation in *Blautia* sp. KLE, respectively. Furthermore, gallic acid metabolism region and its metabolism related enzyme (gallate decarboxylase) encoding genes *lpdB* (gene-V4W90_RS06850; K22959), *lpdC* (gene-V4W90_RS06855; K22959), and *lpdD* (gene-V4W90_RS06845; K22960) were also detected in MBBL6 genome. The secondary metabolite Type III polyketide synthases (T3PKS) in MBBL6 were found to share similarities with two known gene clusters from MIBiG 3.1: fusaricidin B (25% similarity) and xantholipin (4% similarity). The RiPP-like secondary metabolite region in MBBL6 possessed a core biosynthetic gene for a class II bacteriocin, but no similar known gene clusters were identified.

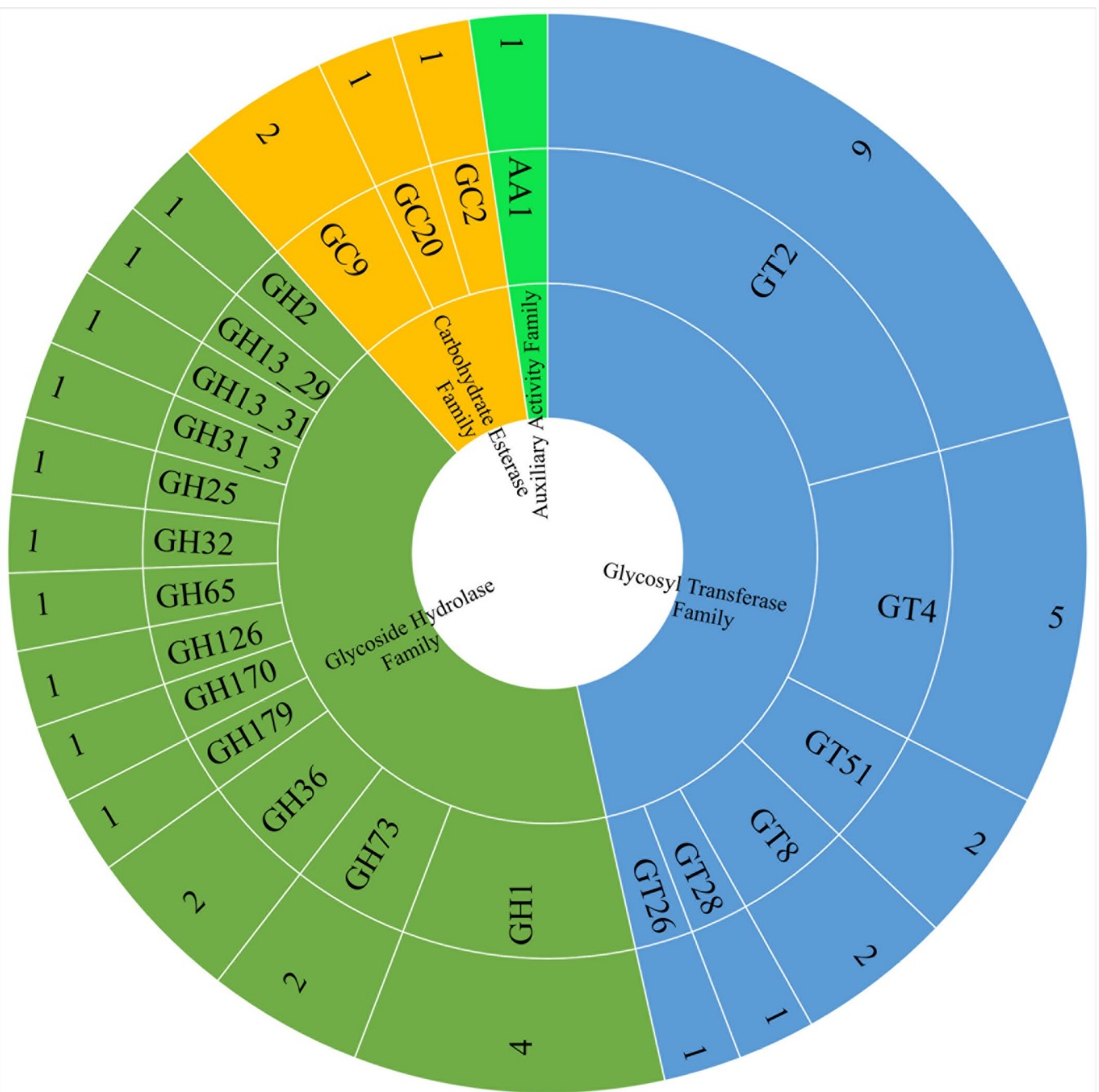

**Fig 7. CAZymes distribution in *P. pentosaceus* MBBL6 genome.** CAZymes were assigned by searching against the CAZy database using the dbCAN webserver. Each CAZymes classes found in the genome are represented by different colors as follows: Green color for auxiliary active family, electric blue for glycosyl transferase family, pistachio color for glycoside hydrolase family, and amber color for carbohydrate esterase family. The rings represented (from the inner to outer rings) CAZyme classes, CAZyme families, and the number of genes identified in each family, respectively.

Genome analysis for bacteriocin biosynthetic gene clusters (BBGC) using BAGEL4 [68] web server identified two BBCG regions in the contig 9 (NZ_JAZIFR0100000091.9.AOI_01; 1–13687) and 12 (NZ_JAZIFR0100000081.12.AOI_01; 31718–51868) of the MBBL6 genome (**Fig 9**). In the NZ_JAZIFR0100000091.9.AOI_01 region (contig 9), we identified a single core peptide known as bovicin_255_variant (**Fig 9A**), whereas two core peptides (penocin_A) were

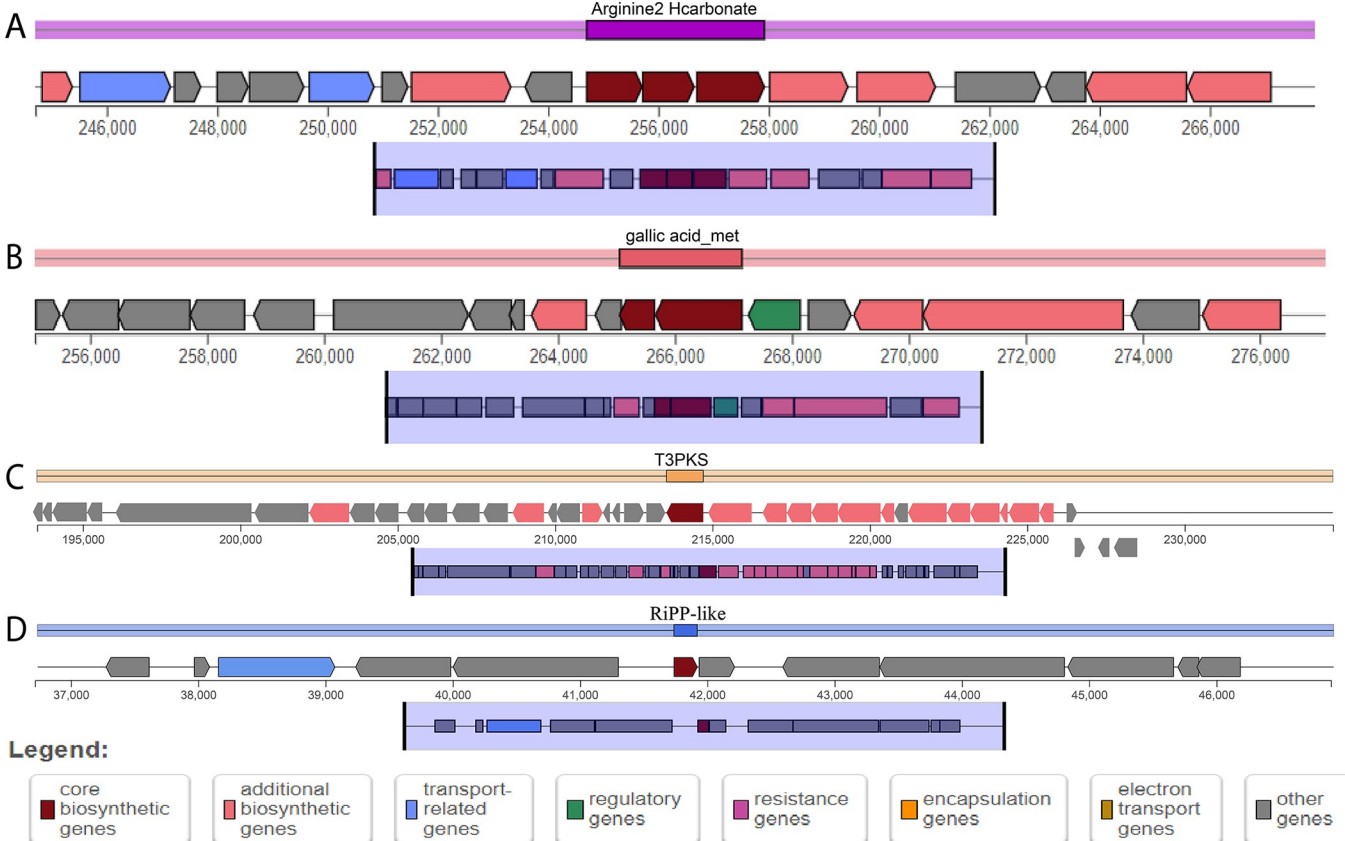

**Fig 8.** Overview of the primary metabolic (A, B) and secondary metabolites (C, D) biosynthesis gene clusters in *P. pentosaceus* MBBL6 genome. (A) Arginine to hydrogen carbonate (Arginine2_Hcarbonate) metabolism, (B) Gallic acid metabolism (Gallic_acid_met), (C) Type III polyketide synthases (T3PKS) and (D) Ribosomally synthesized and post-translationally modified peptide (RiPP)-like regions. Different colors represent genes involved in different functions: Red color for core biosynthesis genes, dark pink color for additional biosynthesis genes, sky blue color for transport-related genes, dark green color for regulatory genes, hot magenta for resistant genes, amber for encapsulation genes, golden brown for electron transport genes, and gray color for other genes.

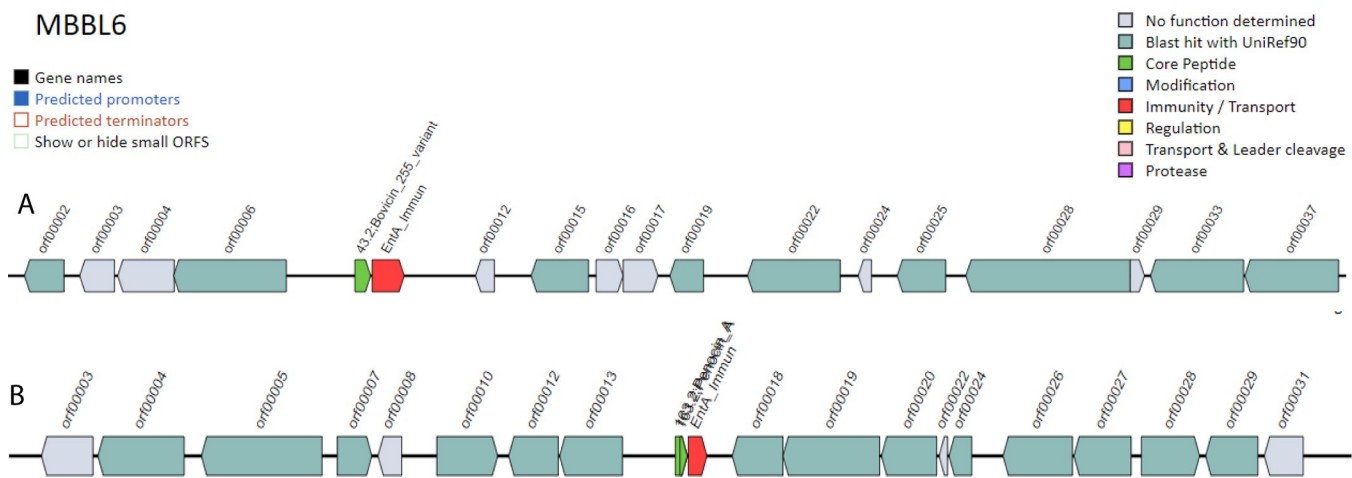

**Fig 9. Overview of bacteriocin producing gene cassettes in *P. pentosaceus* MBBL6 genome.** (A) *bovicin 255* variant, and (B) *penocin A* gene clusters. Interestingly, the immunity/transport gene or protein enterocin A immunity (*entA_immun*) is present just right to the bacteriocin cassette.

identified in contig 12 (NZ_JAZIFR0100000081.12.AOI_01) region (**Fig 9B**). NCBI protein blast confirmed that penocin_A and bovicin_255_variant showed 100% identity with class II bacteriocin and garvicin Q family class II bacteriocin, respectively. Moreover, both regions feature a single enterocin A immunity protein (entA_immun; PF08951) for bacteriocin immunity protein.

## 3.8 *In-vitro* and genomic analyses suggest the safe use of *P. pentosaceus* MBBL6 as a probiotic

*In-vitro* and genomic analyses suggest that MBBL6 is safe as a probiotic, with minimal resistance and no virulence traits. The antibiotic susceptibility test (AST) profile revealed that the MBBL6 isolate was susceptible to the majority (80.0%) of the tested antibiotics, including amikacin, gentamicin, cefepime, ceftriaxone, cefuroxime sodium, imipenem, meropenem, and ciprofloxacin. In contrast, the isolate exhibited resistance to only two antibiotics *i.e.*, nalidixic acid and azithromycin (**S7 Table**). Additionally, genome annotation revealed the presence of only two genes associated with antibiotic resistance (ARGs) namely *lnu*(A) and *erm*(B) in MBBL6 using ResFinder 4.0, suggesting resistance against lincosamide, macrolide and streptogramin (**S8 Table**), and no virulent genes were detected. According to the PathogenFinder [69], which assesses pathogenicity, MBBL6 received a score of 0.196 and aligned with 121 nonpathogenic bacterial families, suggesting that it is unlikely to be a human or animal pathogen. Furthermore, when cultured on blood agar supplemented with 5% sheep blood, the MBBL6 strain exhibited Gamma hemolysis (γ), indicating that it does not cause red blood cell lysis under these conditions (**S2 Fig**). Mobile genetic elements (MGEs), such as prophages, phages, CRISPR systems and plasmids are segments of bacterial DNA that play crucial roles in bacterial evolution, adaptation, and virulence [70,71]. In the MBBL6 genome, four prophage regions were identified including two intact phages, one incomplete phage, and one questionable phage by the PHASTEST web tool (**S3 Fig**). Detailed information about the phage regions is provided in **S9 Table**. Additionally, 32 insertion sequences (IS) were identified in the MBBL6 genome, originating from various genera such as *L. monocytogenes* (Ismo16), *E. coli* (ISEc66), *Mycobacterium bovis* (ISMbov2), *Lactococcus garvieae* (ISLgar1), and *Lactobacillus salivarius* (ISLasa2). Moreover, three CRISPR regions were identified, but no Cas proteins were detected in the MBBL6 genome, as analyzed using the CRISPRcasFinder [58] online server (**S10 Table**).

## 3.9 *P. pentosaceus* MBBL6 exhibits potential antimicrobial efficacy against mastitis pathogens *in-vitro* and *in-vivo*

Our studied strain MBBL6 demonstrates noteworthy antimicrobial activities against two major pathogens namely *S. aureus* D4C4 and *E. coli* G1C5 of bovine mastitis. This was assessed using both *in-vitro* and *in-vivo* assays. The *in-vitro* agar well diffusion technique revealed inhibition zones of 18 mm for *S. aureus* D4C4 and 21 mm for *E. coli* G1C5. These results indicate a substantial inhibitory effect of the MBBL6 isolate against these pathogens. The effectiveness of MBBL6 was further validated in a mouse mastitis model, where it demonstrated notable efficacy in reducing symptoms and controlling infection caused by *S. aureus* D4C4 and *E. coli* G1C5 (**S4A and S4B Fig**). The potential of MBBL6 as a therapeutic candidate was confirmed through its significant antimicrobial activity observed in the *in-vivo* model (**S4C–S4E Fig**). In histopathological examination, mastitis induced mice treated with MBBl6 (Group I) showed no pathological changes in their mammary glands (**S4C–S4E Fig**) and revealed normal epithelial texture in the mammary glands (**S4H Fig**) and colon tissues (**S4K Fig**). In contrast, the untreated mastitis-induced mice in the control group (Group II, **S4F Fig**) exhibited significant histopathological alterations in their mammary gland (**S4G and S4I Fig**) and colon (**S4J and**

**S4L Fig**) tissues. These changes included broken lobules of the mammary gland, damaged acini, destroyed or degenerated epithelial cells, with inflammatory cells, including macrophages, neutrophils, and blood cells detected in the mammary lobule, supporting connective tissue and lining of the epithelium (**S4G and S4I Fig**). Furthermore, inflammatory changes in the colon tissues of the challenged mice included severe infiltration of inflammatory cells (e.g., macrophages, neutrophils, and blood cells) into mucosa and submucosa, degenerated mucosal structure (epithelial necrosis, extension of the subepithelial space, and structural damage of villi) (**S4J and S4L Fig**).

### 3.10 *P. pentosaceus* MBBL6 demonstrates potential antimicrobial efficacy against bovine mastitis pathogens *in-silico*

To investigate the inhibitory ability of MBBL6 against the bovine mastitis pathogens *S. aureus* D4C4 and *E. coli* G1C5, we conducted an in-depth core genomic analysis. Our core genomic analysis identified 56 shared proteins between two pathogens. Further investigation revealed that 26 of these proteins are essential for bacterial survival. Additionally, we identified two potential bacteriocins in MBBL6, bovicin 255 and penocin A. We hypothesized that these bacteriocins might interact with the common essential proteins, disrupting the pathogens' survival mechanisms. This hypothesis was based on the observed inhibitory effects of MBBL6 both *in-vitro* and *in-vivo*. Screening of bacteriocins against the 26 common essential proteins revealed that bovicin strongly interacts with Rho proteins, which function as transcription termination factors (**Fig 10A**). A strong protein-protein interaction was observed between 12 amino acids of bovicin and 11 amino acids of the Rho protein (**Fig 10B**). These interactions are robust, leading to a rigid binding with the Rho protein and disrupting its function in transcription termination. A 200 ns simulation (MDS) demonstrated that the resulting protein-ligand complex is energetically favorable and stable, with an overall Root Mean Square Deviation (RMSD) value of up to 3.5 Å, compared to 4.5 Å (with higher fluctuation) for the Rho protein alone (**Fig 10C**). This indicates that the complex formation between bovicin and Rho results in energetically favorable interactions (**Fig 10C** and **S1 File**).

## 4. Discussion

Probiotics have been associated with a multitude of beneficial effects on host health, including their ability to positively modulate the gut microbiome and impart a wide range of health benefits [72]. *P. pentosaceus* is a bacterium well-known for its probiotic properties, and roles in improving gut health, antimicrobial production, and enhancing fermented food quality [5,6,73]. By whole genome sequencing and comprehensive analysis of *P. pentosaceus* MBBL6, a novel strain isolated from healthy cow milk, we provide data showing its potential as an agent of wide therapeutic applications, bioremediation, and biopreservation. The draft genome of MBBL6 measured 1.84 Mbp, with a genome coverage and GC content of 128.7x and 37.3%, respectively, consistent with previously sequenced *P. pentosaceus* genomes [15,16]. ANI analysis indicated that the MBBL6 genome clustered within the same branch as eight other strains when compared against nineteen strains of *P. pentosaceus* from the NCBI database. The high ANI values observed among all *P. pentosaceus* strains, including MBBL6, suggest limited genetic diversity and potential adaptability to diverse ecological niches [6,74]. The evolutionary analysis of 20 *P. pentosaceus* strains showed clustering of MBBL6 with *P. pentosaceus* strains MBBL4 (cow milk) [6], SRCM217654 (cow feces; NCBI accession: GCF_028656215.1), and DSPZPP1 (fermented sausages) [74], suggesting they are closely related. In this investigation, orthogroups containing all studied genomes are termed core orthogroups, while all other orthogroups are classified as accessory orthogroups. The genome of MBBL6 includes 1,728

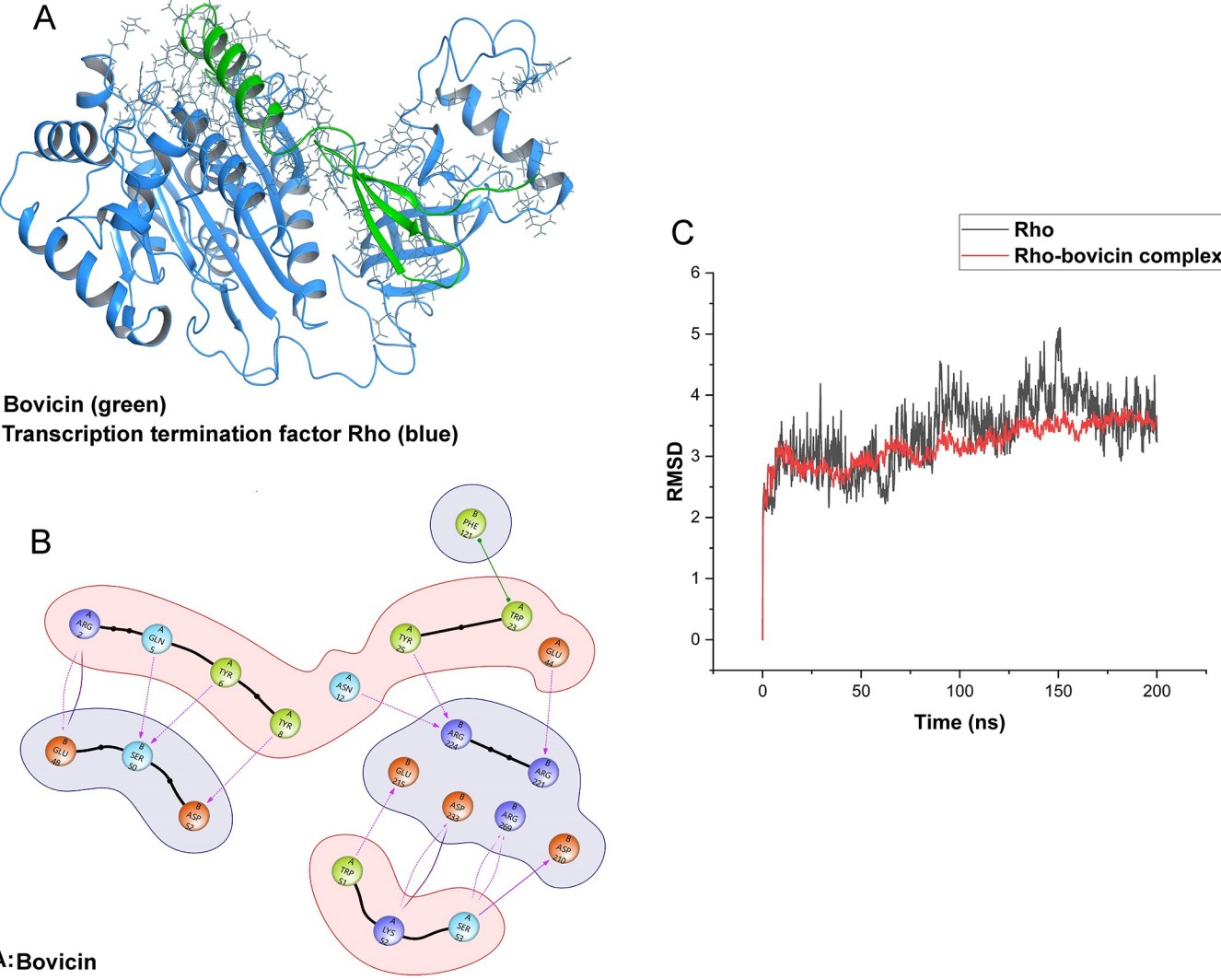

**Fig 10. Interaction between bovicin and the Rho factor.** (A) Molecular screening revealed strong interactions between bovicin and the transcription termination factor Rho. (B) The interaction diagram shows the amino acids involved in the protein-protein interaction. (C) The Root Mean Square Deviation (RMSD) curve confirms the stability of the interaction between bovicin and the Rho factor.

homologous genes, which are primarily involved in essential maintenance functions crucial for the growth and survival of the species[15,16].

One of the hallmark findings of this study may be the identification of genes for probiotic traits such as heat shock, surface adhesion, acid tolerance, antioxidants, and an immune function related gene in MBBL6 genome. These genes are crucial for mitigating the pleiotropic effects of heat stress and play essential roles in the reconstitution and repair of damaged proteins [75]. The protein encoded genes *groEL*, *groES*, *recA*, *dnaK*, *celA*, *celB*, *celC* and *clp* complex, present in the MBBL6, are also protective for bacteria against acidic stress condition [76,77]. The generic stress response related genes (*dnaK*, *groEL*) play critical roles to regulate or restore cellular component functions in the cell by refolding chaperonin complexes capacity [78]. The probiotic organisms generally have good adhesion ability by which they can compete with pathogens for the host binding sites to element the pathogens [5,73]. Recently, it has been

reported that the surface-associated protein elongation factor-Tu plays a role in gut immunity and homeostasis by binding to intestinal cells and inhibiting apoptosis [79]. Among the identified antioxidant genes in MBBL6 genome, thioredoxins encoding genes (*trxA* and *trxB*) play critical roles in defense against oxidative stress by regulating intracellular dithiol/disulfide equilibrium [80]. The gene encoding methionine sulfoxide reductase (*msrC*) restore the oxidized methionine residues [81], glutathione reductase (*gor*) regulate glutathione [82], and 6-phosphogluconate dehydrogenase (*gntZ*) produce NADPH, essential for protecting the cell against oxidative stress [83]. The *metK* gene detected in MBBL6 is associated with its immunity, particularly through DNA methylation and play a critical role in defense mechanisms against foreign DNA [84]. The draft genome of MBBL6 includes genes responsible for the biosynthesis of vitamins and amino acids, making it a robust candidate for probiotic use. Notably, it has a full set of genes for riboflavin biosynthesis, which can reduce gut inflammation through immunomodulation and help maintain gut microbiota homeostasis [85]. Riboflavin deficiency is linked to various health conditions, including migraines, anemia, cataracts, oxidative stress, diabetes mellitus, hypertension, and even cancer [86]. We detected two important lactose metabolism related genes *lacZ* and *lacD* in MBBL6 which can breakdown lactose into simpler sugar forms glucose and D-galactose. Lactose fermentation character is generally absent in naturally occurring *Pediococcus* spp. [87], contrasting with our findings. However, the lactose fermentation capacity observed in MBBL6 is consistent with recent research findings that have documented similar capabilities in other strains of *P. pentosaceus* isolated from diverse sources [15,16]. This underscores the variability within the species and strengthens the understanding of its metabolic versatility across different ecological niches [5,73]. Furthermore, the biochemical tests for MBBL6 showed that the strain can ferment several sugars, including maltose, mannitol, mannose, sucrose, trehalose, and N-acetyl glucosamine. It also has notable enzyme activities, such as arginine utilization, urease production, and beta-glucosidase activity. The Kligler Iron Agar (KIA) test indicated that MBBL6 can ferment both glucose and lactose, as evidenced by the yellow color in both the slant and butt [88]. Conversely, the negative results for alkaline phosphatase, beta-glucuronidase, beta-galactosidase, indole, methyl red, Voges-Proskauer, and citrate tests align with the typical biochemical profiles of *Pediococcus* species [3]. Another key finding of this study is the prediction of carbohydrate-active enzymes (CAZymes). Analysis of the MBBL6 genome revealed 43 CAZymes, categorized into GTs, GHs, CEs, and AAs classes, primarily involved in carbohydrate synthesis and hydrolysis during fermentation. CAZymes play a crucial role in carbohydrate biotransformation (from synthesis to metabolism) process that are essential for every life [89]. In the MBBL6 genome, GT2 and GT4 predominated among all GT classes (70%, 14 out of 20 GTs), involved in synthesizing sucrose, cellulose, chitin, and various glycosyltransferases [90]. The genome also harbors 13 different GH families, facilitating glycosidic bond cleavage in glucose, starch, sucrose, and other carbohydrates. Notably, genes encoding β-glucosidase (GH1), β-galactosidase (GH2), and α-glucosidase (GH13_31) were identified, crucial for lactose, sucrose, and oligosaccharide metabolism [91]. Lysozymes (GH25 and GH73), which serve as a primary defense against bacterial infections, were also identified in MBBL6 genome. These enzymes act as a catalyst for hydrolysis of the β-(1–4) linkages between N-acetylglucosamine and N-acetylmuramic acid residues in the peptidoglycan of bacterial cell walls, which have antimicrobial activity, especially against Gram-positive bacteria [92]. Lactose is a primary source of carbohydrate of the milk and milk derived foods which is also important for fermentation of dairy products. During fermentation, bacteria use lactose to form lactic acid and lower the pH which aids in food preservation by inhibition of food spoilage organisms [93].

Remarkably, MBBL6 harbors genetic regions responsible for gallic acid metabolism and associated genes, which are known for their diverse biological activities including

biopreservation and potential health benefits [19,20]. These specific genes in MBBL6 facilitate the breakdown of gallic acid, enhancing its absorption and bioavailability in the host. This mechanism ultimately contributes to maximizing the therapeutic benefits of gallic acid and promoting better gut health. Furthermore, MBBL6 contains genes encoding secondary metabolites, including T3PKS for synthesizing polyketide compounds used in various drug formulations such as anti-cancer drugs, antibiotics, and cholesterol-lowering agents [94]. Furthermore, the RiPP-like region in MBBL6 is equipped with a core biosynthetic gene for class II bacteriocins, which contributes significantly to its broad-spectrum antimicrobial activity against pathogens [95]. A notable discovery of this study is the detection of bacteriocin-producing gene clusters, such as *penocin A* and *bovicin_255*, in the MBBL6 genome. Penocin A, a class II pediocin bacteriocin, exhibits broad-spectrum antimicrobial activity against pathogens namely *Listeria*, *Enterococcus*, and *Clostridium* species [10]. Additionally, bovicin is capable of inhibiting or killing the major bacterial pathogens causing bovine mastitis [96].

In terms of antibiotic susceptibility, MBBL6 was effective against most of the tested antibiotics, including amikacin, gentamicin, and imipenem, among others. However, it showed resistance to nalidixic acid and azithromycin. The MBBL6 genome contained only two ARGs, *lnu*(A) and erm(B), which are associated with resistance to lincosamides, macrolides, and streptogramins. The presence of these ARGs suggests MBBL6 may have reduced susceptibility to specific antibiotics. However, the practical impact depends on gene expression levels and the overall resistance profile of MBBL6 [15,97]. The absence of other important ARGs indicates it may remain susceptible to many antibiotics, making it a potential candidate for probiotics. The MBBL6 strain exhibited nontoxic γ-hemolysis activity (growth without a zone of hemolysis) when cultured on blood agar plates. Furthermore, no virulence genes were found, and the low pathogenicity score (0.196) indicates that the genome suggests non-pathogenic characteristics suitable for safe use in biological applications. MGEs like plasmids, prophages, and IS are vital as they can transfer ARGs and virulence genes between species, leading to the emergence of superbugs [98]. Although two linear plasmid regions, four phage regions, and three CRISPR regions were identified in the MBBL6 genome, no ARGs or virulence genes were detected within these regions, suggesting its safety as a promising probiotic. The antimicrobial efficacy of MBBL6 was also noteworthy. In laboratory tests, it inhibited the growth of *S. aureus* D4C4 and *E. coli* G1C5, both of which are major pathogens in bovine mastitis [99]. Recently, another strain of *P. pentosaceus*, ENM104, demonstrated probiotic potential through antimicrobial activity against *P. aeruginosa*, *K. pneumoniae*, and *A. baumannii* [11]. While this strain showed no efficacy against *E. coli*, our findings align with those of Zaghloul and Halfawy [13] and Thao et al. [12], who reported that various *P. pentosaceus* strains effectively inhibited *E. coli*, *S. aureus*, *B. subtilis*, *K. pneumoniae*, *P. aeruginosa*, and *E. faecalis*. The *in-vivo* tests through a mouse mastitis model demonstrated that *P. pentosaceus* MBBL6 effectively suppressed mastitis pathogens such as *S. aureus* D4C4 and *E. coli* G1C5, supporting its therapeutic potential. The strain exhibited significant antimicrobial activity, reducing infection severity and alleviating inflammation in the model. These results underscore the promising role of MBBL6 as a potential probiotic treatment for mastitis, further validating its efficacy observed in *in-vitro* studies.

One of the key findings of this study was the elucidation of a possible molecular mechanism by which the MBBL6 can inhibit the growth of *S. aureus* D4C4 and *E. coli* G1C5. Our analysis revealed that bovicin strongly interacts with the Rho factor, a protein involved in terminating transcription and essential for playing a critical role in regulating gene expression and ensuring proper RNA synthesis [100]. The binding of bovicin to these proteins suggests a potential mechanism by which MBBL6 exerts its inhibitory effects, potentially disrupting the transcription termination process and thereby hindering the survival and proliferation of *S. aureus*

D4C4 and *E. coli* G1C5. This robust interaction between bovicin and Rho protein may be the primary reason behind the inhibitory effect, a discovery that we reported in our current investigation. These results suggest that MBBL6 could be considered as a promising candidate for safe use as probiotic or therapeutic agent for managing mastitis in lactating mammals and other biotechnological applications.

## 5. Conclusions

This study provides a comprehensive genomic analysis of the non-hemolytic *P. pentosaceus* MBBL6 strain, isolated from the milk of healthy cows, to evaluate its probiotic potential. The genome sequencing of MBBL6 uncovered a range of functional genes associated with stress tolerance, surface adhesion, immunity, antioxidant properties, carbohydrate utilization, and vitamin B biosynthesis. Notably, we identified important bacteriocin gene clusters, including those encoding *bovicin 255* and *penocin A*, as well as gene clusters involved in secondary metabolite production and primary metabolic pathways, particularly those related to gallic acid metabolism. The genomic analysis revealed no transmissible antibiotic resistance genes or virulence factors in prophage or plasmids, highlighting the strain's safety profile. Furthermore, MBBL6 showed susceptibility to various antibiotics in *in-vitro* tests, and effectively suppressed mastitis-causing pathogens (e.g., *S. aureus* and *E. coli*) both in *in-vitro* and *in-vivo* mouse mastitis model. Its bacteriocin, specifically bovicin, displayed the capability to interfere with crucial proteins such as the Rho factor in these pathogens in an *in-silico* core genomic experiment (**S1 Graphical** abstract). These findings underscore the strain's robust antimicrobial potentials. Overall, *P. pentosaceus* MBBL6 emerges as a highly promising candidate for the development of innovative biotherapeutics for tackling mastitis in the dairy animals, biopreservation strategies, and bioremediation applications. The novel genomic insights gained from this study offer a valuable foundation for further research and application of this strain in various biotechnological and health-related fields.

## Supporting information

**S1 File. Molecular dynamic simulation of bacteriocin (bovicin) and Rho protein of mastitis causing pathogens (*S. aureus* D4C4 and *E. coli* G1C5).**
(MP4)

**S1 Fig. Clustered heatmap of average nucleotide identity (ANI) values between the study strain (MBBL6) and 19 reference genomes of the *P. pentosaceus*.** The numbers represent the ANI values (%) between two genome sequences. The proposed species cut-off boundary is above 98%, demonstrating identity with these strains. Study genome is highlighted in red box.
(TIFF)

**S2 Fig. Growth of *P. pentosaceus* on blood agar with 5% (v/v) sheep blood (Gamma hemolysis).**
(TIF)

**S3 Fig. Identification of prophage region of *P. pentosaceus* MBBL6 genome by PHASTEST.** The *P. pentosaceus* MBBL6 genome harbors two intact (faded green color), one incomplete (faded red color), and one questionable (light yellow color) prophage regions. Circles (from inside to outside) 1 and 2 (GC content and GC skew), circle 3 (phage regions), circles 4 and 5 (phage genes), and circles 6 and 7 (bacterial genes).
(TIF)

**S4 Fig. *P. pentosaceus* MBBL6 exhibited potential antimicrobial efficacy against bovine mastitis pathogens in *in-vivo* mouse mastitis model experiment.** Pathophysiological changes in the mammary glands and gut (colon) of mice after experimental challenge with two major pathogens (*S. aureus* D4C4 and *E. coli* G1C5) of bovine mastitis. The mice were challenged at Day 18 of their gestation. (A–B) Two groups of challenged mice. (C) Mouse with induced mastitis showing gross pathological changes in mammary glands (swollen, red and inflamed mammary glands) seven days after challenge. (D) *P. pentosaceus* MBBL6 treated mouse (at day 7 of treatment) showing mild inflammatory changes in the mammary glands. (E) *P. pentosaceus* MBBL6 treated mouse (at day 14 of treatment) showing no visible inflammatory changes in the mammary glands. (F) Mouse with induced mastitis receiving only placebo treatment (control). (G–L) Representative photomicrographs of mammary and colon tissues after haematoxylin and eosin (HE) staining. (G and I) The lesions in the mammary alveoli are characterized by a central area of necrosis, broken lobules, damaged acini and destroyed epithelial cells, with large numbers of inflammatory cells, predominantly neutrophils (dark bluish) and macrophages (red) in the mammary lobule, supporting connective tissue and lining of the epithelium. (H) Representative photomicrographs of healthy mammary gland showing no visible inflammatory changes in the epithelium and alveoli after HE staining. (J and L) Representative photomicrographs of colon (crypts, lamina propria, muscularis mucosae and submucosa) tissue after HE staining. Inflammatory changes include moderate to severe inflammatory cell infiltration into mucosa and submucosa, disorder in mucosal structure: Epithelial necrosis, extension of the subepithelial space, and structural damage of villi. (K) Representative photomicrographs of colon tissue of healthy mouse after HE staining where no inflammatory changes were observed. Inflammatory changes in the mammary glands of the mice are mouse are highlighted in purple arrows. Scale bars: 50 µm. (TIF)

**S1 Table. BioSample ID, isolation source, and country of origin of 19 *P. pentosaceus* strains.**
(DOCX)

**S2 Table. Carbohydrate fermentation and enzyme activities of the *P. pentosaceus* MBBL6 isolate.**
(DOCX)

**S3 Table. Prediction of RNAs in *P. pentosaceus* MBBL6.**
(DOCX)

**S4 Table. Carbohydrate metabolism related protein encoding genes predicted in *P. pentosaceus* MBBL6.**
(DOCX)

**S5 Table. Probiotic and vitamin B biosynthesis related genes predicted in *P. pentosaceus* MBBL6.**
(DOCX)

**S6 Table. Prediction of primary metabolic and secondary metabolite biosynthesis gene clusters in *P. pentosaceus* MBBL6.**
(DOCX)

**S7 Table. Antibiotic susceptibility results from culture plate: R = Resistance, S = Susceptible, I = intermediate AST results.** The third column in the right column

represents zone of inhibition (ZoI).
(DOCX)

**S8 Table. Prediction of antibiotic resistance genes in *P. pentosaceus* MBBL6.**
(DOCX)

**S9 Table. Details of prophage regions of *P. pentosaceus* MBBL6.**
(DOCX)

**S10 Table. Prediction of CRISPER/Cas in *P. pentosaceus* MBBL6.**
(DOCX)

**S1 Graphical abstract. Schematic representation of genomic and probiotic insights into *Pediococcus pentosaceus* MBBL6, isolated from healthy cow milk.** The study involved sequencing and annotating the genome of *P. pentosaceus* MBBL6 utilizing advanced bioinformatics tools to elucidate its metabolic pathways, carbohydrate-active enzymes, bacteriocin gene clusters, and secondary metabolite biosynthetic gene clusters. Antimicrobial susceptibility testing of MBBL6 was performed against mastitis-causing pathogens, including *S. aureus* and *E. coli*, through both *in-vitro* and *in-vivo* trials. Additionally, an *in-silico* analysis of the core genome was conducted to gain deeper insights into the possible molecular mechanisms by which *P. pentosaceus* MBBL6 inhibits the growth of these mastitis-causing pathogens. he results indicate *P. pentosaceus* MBBL6 is a promising probiotic and/or therapeutic agent for mastitis management and biotechnological applications.
(TIFF)

## Acknowledgments

The authors express their gratitude to Dr. Salma Akter, Associate Professor at the Department of Microbiology, Jahangirnagar University, Savar, Dhaka 1342, Bangladesh, for her assistance with biochemical analysis.

## Author Contributions

**Conceptualization:** Md. Morshedur Rahman, Tofazzal Islam, M. Nazmul Hoque.

**Data curation:** Md. Morshedur Rahman, Naim Siddique, Soharth Hasnat, M. Nazmul Hoque.

**Formal analysis:** Md. Morshedur Rahman, Naim Siddique, Soharth Hasnat, M. Nazmul Hoque.

**Funding acquisition:** Ziban Chandra Das, M. Nazmul Hoque.

**Investigation:** Md. Morshedur Rahman, Md. Tanvir Rahman, Mustafizur Rahman, Munirul Alam, Ziban Chandra Das, M. Nazmul Hoque.

**Methodology:** Md. Morshedur Rahman, Naim Siddique, Soharth Hasnat, Ziban Chandra Das, M. Nazmul Hoque.

**Project administration:** Md. Tanvir Rahman, Mustafizur Rahman, Munirul Alam, Ziban Chandra Das, Tofazzal Islam, M. Nazmul Hoque.

**Resources:** Mustafizur Rahman, Munirul Alam, Ziban Chandra Das, Tofazzal Islam, M. Nazmul Hoque.

**Software:** Md. Morshedur Rahman, Soharth Hasnat, M. Nazmul Hoque.

**Supervision:** Md. Tanvir Rahman, Mustafizur Rahman, Munirul Alam, Ziban Chandra Das, Tofazzal Islam, M. Nazmul Hoque.

**Validation:** Soharth Hasnat, Md. Tanvir Rahman, Mustafizur Rahman, Munirul Alam, Ziban Chandra Das, M. Nazmul Hoque.

**Visualization:** Md. Morshedur Rahman, Soharth Hasnat.

**Writing – original draft:** Md. Morshedur Rahman, Naim Siddique, Soharth Hasnat.

**Writing – review & editing:** Md. Tanvir Rahman, Mustafizur Rahman, Munirul Alam, Ziban Chandra Das, Tofazzal Islam, M. Nazmul Hoque.

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
