## [Decision Letter · Decision Letter 0]

13 Nov 2024

PONE-D-24-42392Genomic Insights into the Probiotic Potential and Genes Linked to Gallic Acid Metabolism in Pediococcus pentosaceus MBBL6 Isolated from Healthy Cow MilkPLOS ONE

Dear Dr. Hoque,

Thank you for submitting your manuscript to PLOS ONE. After careful consideration, we feel that it has merit but does not fully meet PLOS ONE’s publication criteria as it currently stands. Therefore, we invite you to submit a revised version of the manuscript that addresses the points raised during the review process.

I am pleased to offer conditional acceptance on the basis that the following minor comments from Reviewer 2 and Reviewer 3 are addressed. Comments made by Reviewer 3 can be pointed out on the manuscript (mention line numbers) in the submission of your rebuttal. In a case where this is not possible, clarify the according to the Reviewer’s comments.

We look forward to receiving your revised manuscript.

Kind regards,

Edwin Hlangwani

Academic Editor

PLOS ONE

2. Thank you for your submission to PLOS ONE. We note that your study design may include death of a regulated animal as a likely outcome or planned experimental endpoint. At this time, we request that you please report additional details in your Methods section regarding animal care and use for the survival study, as per our editorial guidelines (http://journals.plos.org/plosone/s/submission-guidelines#loc-humane-endpoints).     

For easy reference, we have attached a checklist that may be relevant for your submission. Please complete all items on the checklist at the following link:   http://journals.plos.org/plosone/s/file?id=bb1d/plos-one-humane-endpoints-checklist.docx        

Please upload the completed checklist as file type “Other” when resubmitting your manuscript. This document is for internal journal use only and will not be published if your article is accepted. We very much appreciate your attention to these requests and support of improved reporting standards in PLOS ONE submissions.

3. To comply with PLOS ONE submissions requirements, in your Methods section, please provide additional information regarding the experiments involving animals and ensure you have included details on (1) methods of sacrifice, (2) methods of anesthesia and/or analgesia, and (3) efforts to alleviate suffering.

4. Please note that PLOS ONE has specific guidelines on code sharing for submissions in which author-generated code underpins the findings in the manuscript. In these cases, we expect all author-generated code to be made available without restrictions upon publication of the work. Please review our guidelines at https://journals.plos.org/plosone/s/materials-and-software-sharing#loc-sharing-code and ensure that your code is shared in a way that follows best practice and facilitates reproducibility and reuse.

“This work was supported by research grants from the BSMRAU Innovation Fund 2023-2024 (Grant No. 14) and the Research Management Wing (RMW), BSMRAU (Grant No. 18, FY 2023-2025), Bangladesh.”

6. We are unable to open your Supporting Information file [Supplementary Information.rar]. Please kindly revise as necessary and re-upload.

Reviewers' comments:

Reviewer's Responses to Questions

**Comments to the Author**

1. Is the manuscript technically sound, and do the data support the conclusions?

Reviewer #1: Yes

Reviewer #2: Yes

Reviewer #3: Yes

2. Has the statistical analysis been performed appropriately and rigorously? 

Reviewer #1: N/A

Reviewer #2: Yes

Reviewer #3: Yes

3. Have the authors made all data underlying the findings in their manuscript fully available?

Reviewer #1: Yes

Reviewer #2: Yes

Reviewer #3: Yes

4. Is the manuscript presented in an intelligible fashion and written in standard English?

Reviewer #1: Yes

Reviewer #2: Yes

Reviewer #3: Yes

5. Review Comments to the Author

Reviewer #1: I commend the comprehensive investigation of P.pentosaceus MBBL6 and its potential as a probiotic. Your findings are outstanding. The paper is well-structured, with a logical flow that guides the reader through the research objectives, methodology, and findings.

Reviewer #2: The manuscript demonstrates a whole genome analysis of P. pentosaceus MBBL6 (isolated from healthy cow's milk) to study its probiotic potential. The study presented is a comprehensive one that shows the involved metabolic pathways, carbohydrate-active enzymes, bacteriocin gene clusters, and secondary metabolite biosynthetic gene clusters. The manuscript could do a better job of describing the strain-specific genes (and associated pathways) to emphasize the significance of MBBL6, based on comparison with closely related genomes. Next, the manuscript could also discuss how the results compare with other strains (for example, Kompramool S., et al, 2024) that have also demonstrated antimicrobial activity. I recommend that the manuscript be published once these abovementioned have been addressed. Overall, the distinct or unique molecular functions of the strain based on a comparative analysis could be discussed further.

Reviewer #3: 1). What are the main reasons for selecting Pediococcus pentosaceus MBBL6 isolated from healthy cow milk at BSMRAU dairy farm, Gazipur, Bangladesh, as the study subject?

2). What are the main objectives of the comprehensive genomic analysis of P. pentosaceus MBBL6 in this study?

3). What methods were used in this study to evaluate the probiotic and antimicrobial potential of P. pentosaceus MBBL6, both in-vitro and in-vivo?

4). What are the main findings related to genes involved in carbohydrate metabolism and vitamin B complex biosynthesis in the genome of P. pentosaceus MBBL6?

5). How do the results of in-vivo tests in a mouse mastitis model support the therapeutic potential of P. pentosaceus MBBL6 in overcoming mastitis pathogens?

6). What are the main conclusions that can be drawn from the genomic analysis of P. pentosaceus MBBL6 related to its applications in therapy, bioremediation, and biopreservation?

7). Use a minimum of the last 5 years of literature, especially research articles.

6. PLOS authors have the option to publish the peer review history of their article (what does this mean?). If published, this will include your full peer review and any attached files.

Reviewer #1: No

Reviewer #2: No

Reviewer #3: No

---

## [Author Response · Author response to Decision Letter 0]

19 Nov 2024

Dear Editor,

Thank you for the editor's decision letter dated on November 13, 2024. Appended to this letter is our point-by-point responses to the comments raised by both reviewers and editors. We would like to take this opportunity to express our sincere thanks to the expert reviewers/editors who identified several areas in our manuscript that were needed corrections as well as modifications. We also would like to cordially thank you for allowing us the change to resubmit a revised version of the manuscript. 

We have revised and updated the manuscript with some modifications as per reviewers’ suggestion. Please find all changes highlighted in RED color fonts in the revised manuscript. We also have provided a clean manuscript for your kind perusal.

Sincerely yours,

M. Nazmul Hoque, PhD

The Corresponding Author

---

## [Editor Report · Decision Letter 1]

9 Dec 2024

Genomic Insights into the Probiotic Potential and Genes Linked to Gallic Acid Metabolism in Pediococcus pentosaceus MBBL6 Isolated from Healthy Cow Milk

PONE-D-24-42392R1

Dear Dr. Hoque,

We’re pleased to inform you that your manuscript has been judged scientifically suitable for publication and will be formally accepted for publication once it meets all outstanding technical requirements.

Kind regards,

Edwin Hlangwani

Academic Editor

PLOS ONE
---

## [Editor Report · Acceptance letter]

13 Dec 2024

PONE-D-24-42392R1 

PLOS ONE

Dear Dr. Hoque, 

I'm pleased to inform you that your manuscript has been deemed suitable for publication in PLOS ONE. Congratulations! Your manuscript is now being handed over to our production team.

Kind regards, 

on behalf of

Dr. Edwin Hlangwani 

Academic Editor

PLOS ONE